# Natural product P57 induces hypothermia through targeting pyridoxal kinase

Ruina Wang[1,11], Lei Xiao [2,11], Jianbo Pan[3,11], Guangsen Bao[1,11], Yunmei Zhu[3], Di Zhu[1], Jun Wang[4], Chengfeng Pei[5], Qinfeng Ma[3], Xian Fu[3], Ziruoyu Wang[1], Mengdi Zhu[5], Guoxiang Wang[2], Ling Gong[2], Qiuping Tong[2], Min Jiang[2], Junchi Hu[3], Miao He [2], Yun Wang[2], Tiejun Li[6], Chunmin Liang [7], Wei Li [8], Chunmei Xia[9], Zengxia Li[1], Dengke K. Ma [10], Minjia Tan [4], Jun Yan Liu [3], Wei Jiang [1] ✉, Cheng Luo [4] ✉, Biao Yu[5] ✉ & Yongjun Dang[3] ✉

Induction of hypothermia during hibernation/torpor enables certain mammals to survive under extreme environmental conditions. However, pharmacological induction of hypothermia in most mammals remains a huge challenge. Here we show that a natural product P57 promptly induces hypothermia and decreases energy expenditure in mice. Mechanistically, P57 inhibits the kinase activity of pyridoxal kinase (PDXK), a key metabolic enzyme of vitamin B6 catalyzing phosphorylation of pyridoxal (PL), resulting in the accumulation of PL in hypothalamus to cause hypothermia. The hypothermia induced by P57 is significantly blunted in the mice with knockout of PDXK in the preoptic area (POA) of hypothalamus. We further found that P57 and PL have consistent effects on gene expression regulation in hypothalamus, and they may activate medial preoptic area (MPA) neurons in POA to induce hypothermia. Taken together, our findings demonstrate that P57 has a potential application in therapeutic hypothermia through regulation of vitamin B6 metabolism and PDXK serves as a previously unknown target of P57 in thermoregulation. In addition, P57 may serve as a chemical probe for exploring the neuron circuitry related to hypothermia state in mice.

Many mammals enter hibernation/torpor in adaption to extreme environmental conditions. Hibernation or torpor is characterized by pronounced temporal reductions in body temperature and energy expenditure. Hibernation-capable mammals can develop resilience to ischemic stress and hypothermia in a therapeutic setting can protect human tissues from cell death in many neurovascular and cardiovascular ischemic diseases[1–8]. Whether and how small molecules pharmacologically induce hypothermia remains largely unknown. Eight classes of agents have been reported to have hypothermic and neuroprotective effects in preclinical ischemic models, including cannabinoids[9–12], transient receptor potential vanilloid 1 (TRPV1) agonists[13,14], opioid receptors agonists[15–20], neurotensin[21–25] and thyroxine derivatives[26,27]. However, these

compounds are rarely used in clinics given undefined clinical efficacy, unclear mechanisms of action in neuroprotection and multiple side effects. It has been known that body temperature is tightly controlled by the preoptic area (POA) in hypothalamus[28–31]. POA can be further divided into several nuclei, such as medial preoptic area (MPA) and ventrolateral preoptic nucleus (vLPO). When the body faces a temperature challenge, the temperature information is sensed by primary sensory ganglia (or trigeminal ganglia), and then passed through the dorsal horn of the spinal cord to the POA of the hypothalamus. Neurons in POA are activated to transmit signals of body temperature regulation back to peripheral effector organs to regulate thermogenesis homeostasis[32]. The activation of either GABAergic or glutamatergic POA neurons can lead to

hypothermia[33,34]. Specific activation of MPA neurons is also reported to reduce body temperature[35,36]. In addition, it has been reported that the activation of neurons in hypothalamic vLPO, dorsomedial hypothalamus (DMH) and other hypothalamus regions is also involved in the regulation of mammalian body temperature[33,37,38]. But the exact regulatory neurons and their connections have not been understood[32,33]. Therefore, compounds that specifically target temperature-sensitive neurons in hypothalamus may be useful not only in the clinic but also as tools to facilitate the investigation of mechanisms and circuitry involved in the induction of hypothermia. Here, we find that natural product P57 induces hypothermia by targeting pyridoxal kinase (PDXK), a key metabolic enzyme of vitamin B6 catalyzing phosphorylation of pyridoxal (PL), resulting in the accumulation of PL in hypothalamus, and has a great neuroprotective effect in the ischemic stroke model, direct injection of the compound into the POA can lead to similar hypothermia effects.

## Results

### P57 induces a hypothermia and hypometabolism in mice

P57, an oxypregnane steroidal glycoside, is a well-known natural product isolated from *Hoodia gordonii* for its prominent activity of appetite suppression through undefined mechanism[39,40]. Upon completion of total synthesis of P57 (Fig. 1a), we began to study its potential functions in rodents. Serendipitously, we observed that intraperitoneal injection of P57 caused a significant and reversible decrease in body core temperature by Anilogger® core temperature monitoring system. Core temperature was reduced by approximately 2 to 5 °C, lasting for 2 to 5 hours with intraperitoneal administration of P57 at 12.5 mg/kg and 25.0 mg/kg, respectively (Fig. 1b). Mammals maintain body temperature by regulating metabolism, prompting us to investigate the metabolic changes induced by P57. Indeed, $O_2$ consumption, $CO_2$ production, respiratory exchange ratio (RER) and energy expenditure were all significantly decreased in P57-treated mice and returned back to baseline levels after around 7 hours (Fig. 1c-e and Fig.S1a). Same as the effect of natural P57[39,40], P57-treated mice had a substantial decrease in food intake (Fig. 1f). We further assessed the effect of P57 on mouse motor activity, and observed that travel distance in the open-field test and latency to fall in the rotarod test were significantly decreased with administration of P57. Mouse motor behaviors can gradually recover (Fig.S1d, e).

We further administrated 4 consecutive intraperitoneal injection to mice with 25.0 mg/kg of P57 every 3 hours and observed that the core temperature of mice could be as low as 26 °C and the hypothermic state ($T_{core} < 34$ °C) lasted for longer than 30 hours with decreased activity (Fig. 1g). In addition, mean blood pressure and heart rate decreased in 24 h and return to normal level in 120 hours (Fig. S2a, b). Gross histological examination of brain, heart, kidney, liver and muscle did not show obvious change after recovery (Fig. S2c). These results indicated that P57 causes a hypophagic, hypothermic and hypometabolic state in a reversible manner.

Hypothermia has neuroprotective effects in ischemic diseases[41–44]. In a mice ischemic stroke model using middle cerebral artery occlusion (MCAO), P57 treatment obviously decreased infarct volumes, while P57 treatment was counteracted by warming (Vehicle: $18.9 \pm 4.2\%$ vs. P57: $6.1 \pm 6.5\%$ vs. P57 + warming: $15.3 \pm 2.6\%$) (Fig. 1h–j). We also evaluated the effect of P57 in rats and found that the core body temperature in rats was also rapidly reduced from around 37.0 °C to 34.0 °C after P57 administration within 1 h and hypothermia lasted for more than 4 h in a dose-dependent manner (Fig. S3a). P57 treatment also obviously decreased infarct volumes in rat ischemic stroke model (Vehicle: $29.9 \pm 4.4$ % vs. P57: $11.9 \pm 5.7$ %) (Fig. S3b, c). These results indicate that P57 induces hypothermia in rodents and has the protective effect on ischemic injuries in the brain.

### P57 binds with PDXK and inhibits its kinase activity

P57 is rapidly diffused to brain after the intravenous administration[45]. To explore the mechanism of P57, we synthesized a biotinylated probe of P57 (Bio-P57) (Fig. 2a), and identified its putative targets through affinity purification from mouse brain tissue lysates, followed by mass spec analysis of P57-bound proteins (Fig. S4a, Table S1). Among the identified proteins, glutathione S-transferase P1 (GSTP1), pyridoxal kinase (PDXK) and neuromodulin were significantly enriched by Bio-P57. Further validation with immunoblotting showed GSTP1 and PDXK, but not neuromodulin, were sensitive to competition by excess P57 (Fig. 2b, Fig. S4b). GSTP1 was excluded as the potential physiological target of P57 since we did not observe any body temperature change when mice were treated with its inhibitors (ethacrynic acid or celastrol[46]) or agonist (α-angelica lactone), as well as the substrate (glutathione, GSH) of GSTP1 (Fig. S4c–e), leaving PDXK as the only remaining target candidate.

PDXK is responsible for the phosphorylation of vitamin B6, which includes pyridoxal (PL), the major form in vivo, pyridoxamine (PM) and pyridoxine (PN), in the presence of ATP and $Zn^{2+}$ or $Mg^{2+}$. Phosphorylation of vitamin B6 constitutes an essential step in the synthesis of pyridoxine 5′-phosphate (PLP), the active form of vitamin B6 and a cofactor for over 160 enzymes. We further characterized P57 binding to PDXK by a pull-down assay using the purified recombinant PDXK protein and observed that binding of biotinylated P57 to PDXK was sensitive to competition by unmodified P57 (Fig. 2c). Microscale thermophoresis (MST) and surface plasmon resonance (SPR) revealed binding affinity between P57 and PDXK, with dissociation constants of $15.8 \pm 2.9$ µM and 3.5 µM, respectively (Fig. 2d, e).

We next determined whether kinase activity of PDXK was affected by P57. We found that P57 inhibited PDXK enzyme activity in a dose-dependent manner, with $EC_{50}$ at 20.0 to 30.0 µM (Fig. S5a). Kinetic studies with P57 wherein ATP concentration was fixed and concentration of PL was varied showed that Vmax remained constant and Km values increased, which indicated that P57 competitively inhibited PDXK with PL (Fig. 2f, Fig. S5b). Similar experiments wherein PL concentration was fixed and the concentrations of ATP were varied showed that both $K_m$ and $V_{max}$ values decreased, indicating that P57 uncompetitively inhibits PDXK with respect to ATP (Fig. S5c).

To investigate how P57 might bind to PDXK, we used the ligand docking program Schrodinger Glide to dock P57 into the PLP binding pocket of PDXK (3KEU)[47]. The procedure was validated by redocking with a root mean square deviation (RMSD) of 0.125 Å between the docked conformation and co-crystallized pose. P57 could be docked into the PL-binding site in a PL competitive way (Fig. 2g, h). P57 is stabilized by hydrogen bonding interactions with Thr47, Gly232 and Asp235 residues at the active site. To further verify our docking model, we evaluated the inhibitory action of P57 with respect to PL. Compared with the PDXK binding in the absence of PL, the apparent $K_d$ for P57 against PDXK was dramatically reduced with increasing concentration of PL (Fig.S5d). Aglycone fragment of P57 showed a dramatically reduced affinity with PDXK ($K_d = 123.5 \pm 35.4$ µM) (Fig. S5e). Kinetic studies with P57-aglycone wherein PL concentration was fixed and the concentrations of ATP were varied showed that Km remained constant and Vmax values decreased, which indicated that P57-aglycone is a non-competitive inhibitor of PDXK (Fig.S5f). D235 and G232 jointly participate in the stabilization of the conformation at one terminal of P57. Accordingly, the D235A mutation can reduce the binding stability to a certain extent, and partly reducing the inhibitory activity. On the contrary, T47V mutation removes the hydrogen bond anchor on the other end, thereby increasing the flexibility of the tail, resulting in a complete loss of the inhibitory activity (Fig.S5g). Together, these results indicate that PDXK is a target of P57 and its kinase activity is inhibited by P57.

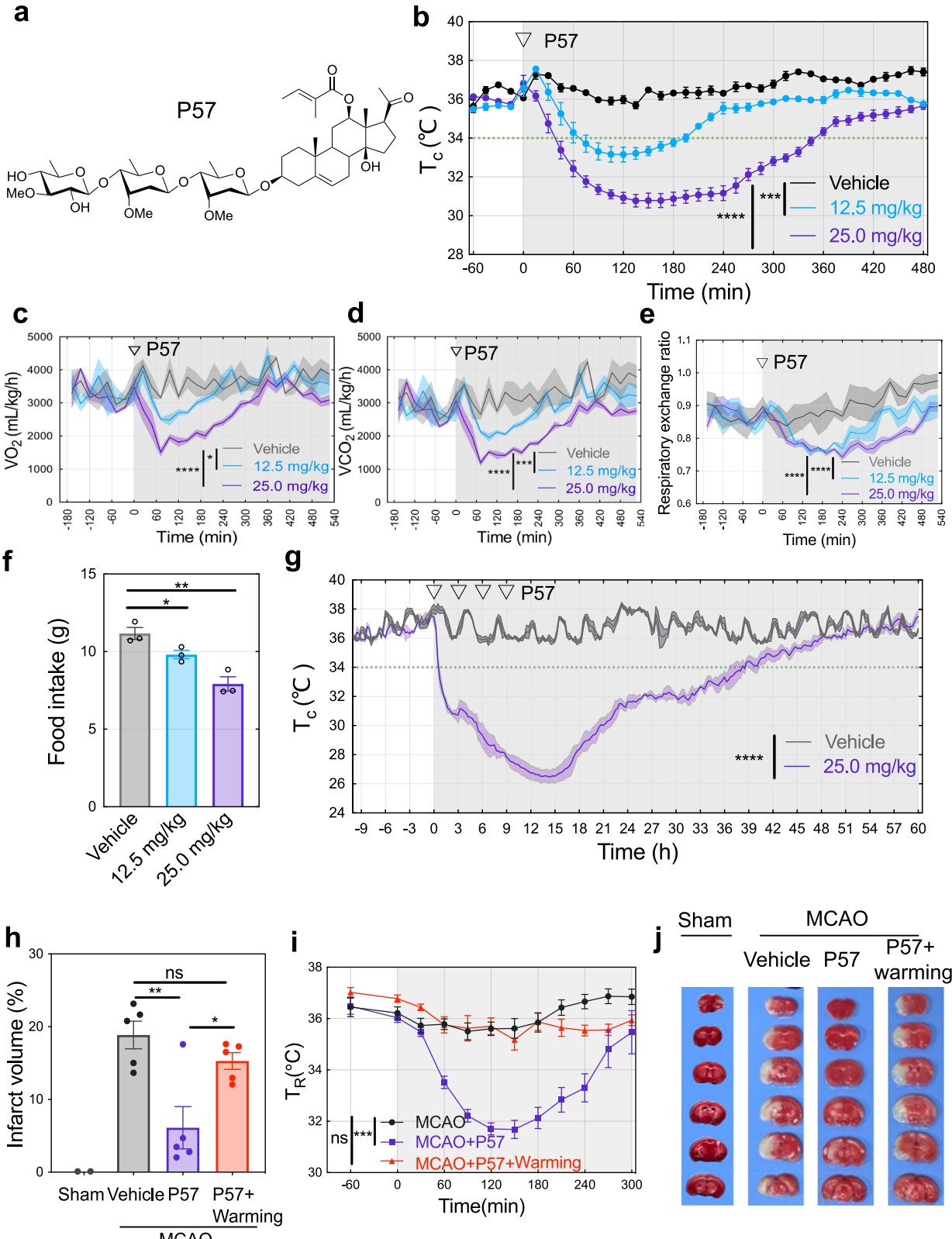

## P57 mainly targets PDXK in hypothalamus to induces hypothermia

To determine whether hypothermia induced by P57 was mediated through PDXK inhibition in vivo, we measured vitamin B6 level in hypothalamus and whole mouse brain after 25.0 mg/kg of P57 treatment using a high-performance liquid chromatography spectrometry. PL level, but not other vitamin B6 forms such as PN, PM and 4-PA, was

dramatically increased in hypothalamus after intraperitoneal administration of P57, but there was no obvious change in whole brain or striatum tissues (Fig. 3a, table S2 and table S3 and Fig.S6a), which suggests the hypothalamus as the effector region of P57-induced hypothermia. Pioneering studies have shown that POA of the hypothalamus, plays a key role in thermoregulation and energy balance[28–31]. We thus locally administered P57 into POA at 0.6 mg/kg and 1.2 mg/kg

**Fig. 1 | P57 induces a reversible hypothermia in mice and has a neuroprotective effect in mouse MCAO model. a** Chemical structure of P57. **b** Core temperature of P57-treated mice. P57 or the vehicle control was injected intraperitoneally into mice at 0 min (arrow). $n = 5$ mice, significant differences between treatments were calculated using two-way analysis of variance (ANOVA). **c** Oxygen consumption level of P57-treated mice. Volume of oxygen ($VO_2$) was measured every 24 min, $n = 3$ mice, two-way ANOVA. **d** Carbon dioxide output level of P57-treated mice. Volume of Carbon dioxide ($VCO_2$) was measured every 24 minutes, $n = 3$ mice, two-way ANOVA. **e** Respiratory exchange ratio of P57-treated mice. Respiratory exchange ratio is the ratio of $VCO_2$ to $VO_2$, $n = 3$ mice, two-way ANOVA. **f** Food intake of P57-treated mice in metabolic chambers. Food intake over 72 hours was measured. $n = 3$ mice, student's t test. **g** Core temperature of mice treated with P57 four times. P57 or the vehicle control was injected intraperitoneally into mice at 0, 3, 6, and 9 h

(arrow). $n = 3$ mice, two-way ANOVA. **h–j** P57 treated-mice after middle cerebral artery occlusion (MCAO). P57 (25.0 mg/kg) or the vehicle was injected intraperitoneally into mouse 2 h after MCAO. Mice were sacrificed at 24 h after drug injection and brain were quickly removed for TTC staining. Core temperature of mice in MCAO + P57+ Warming group maintained by the heating pad. $n = 2$ mice for sham group, $n = 5$ mice for others. **h** Statistic infarct volumes of brain slices from mouse model of MCAO. student's t test. **i** Rectal temperature of mice in MCAO model. student's t test. **j** Representative images of TTC staining of brain slices in MCAO model treated with P57. Experiments in Fig.1 were performed under ambient temperatures at 22–24 °C. Student's t test used was two-sided. All error bars are presented as mean values ± s.e.m. *$P < 0.05$, **$P < 0.01$, ***$P < 0.001$ and ****$P < 0.0001$. Source data are provided as a Source Data file.

respectively and observed rapidly induction of deeper hypothermia in a dose-dependent manner (Fig. 3b, c). However, the core temperature of mice didn't change when P57 was administrated directly into striatum (Fig. S6b, c). Similarly, direct injection of low doses of P57 (150 μg/kg) in bed nucleus of the stria terminalis, lateral division, dorsal part (BSTLD), paraventricular nucleus of hypothalamus (PVH) and lateral preoptic area (LPO) does not result in hypothermia (Fig. S9h–j). Moreover, we expressed Cre-EGFP in POA by injecting $Pdxk^{flox/flox}$ mice with AAV2/9-hSyn-Cre-EGFP-WPRE-pA, the effect of P57 (25.0 mg/kg) on inducing hypothermia was significantly reduced in the mice with knocking out PDXK in POA compared to control group (Fig. 3d–f and Fig.S6d). Moreover, core temperature of mice with PDXK knockout in hypothalamus was lower than the control animals when cold exposure (10 °C) was applied for 24 hours (Fig. 3g). These results suggested that PDXK is validated as the target of P57 to regulate mouse core temperature.

As P57-induced inhibition of PDXK will dramatically increase the PL level in hypothalamus, we directly tested the effect of PL in hypothermia. We observed mouse core body temperature decreased to around 30.0 °C after intraperitoneal administration of PL at 300.0 mg/kg, but other B6 vitamers could not change the core temperature at the same dosage (Fig. 3h and Fig.S6f). In addition, 5.0 mg/kg of PL administrated directly into POA rapidly induced hypothermia (Fig. 3i). Combined intraperitoneal administration of P57 (12.5 mg/kg) and PL (200.0 mg/kg) caused a synergistic reduction of body temperature (Fig. 3j). We also observed that travel distance in the open-field test and latency to fall in the rotarod test were significantly decreased with administration of PL (Fig.S6e, g). These results confirm that P57 inhibits the enzyme activity of PDXK to induce hypothermia through specifically increasing hypothalamus PL level.

### P57 and PL play a similar effect on hypothalamus neurons

To further characterize the state of various cell types in the hypothalamus after administering P57 and PL respectively, and explore the mechanism of regulating core temperature, we performed single-nucleus RNA-sequencing (snRNA-seq) using nuclei dissociated from mouse hypothalamus. C57 wild-type mice were injected with P57 and PL respectively, while mice treated with phosphate buffered saline were included as control. A total of 40,008 single nuclei were analyzed after removing low quality cells and doublets. Unsupervised clustering analysis delineated the dominant neuronal and non-neuronal cells clusters. Glutamatergic and GABAergic neurons, the two most populated clusters ($n = 21411$), are further divided into neuronal subpopulation (24 glutamatergic and 15 GABAergic clusters) (Fig. S7a–f, Fig. S8a). The changes of differentially expressed genes (DEGs) between P57 and PL show a similar pattern (Fig. 4a, b). We also found a significant positive correlation between P57 and PL in the DEGs (Pearson's correlation coefficient 0.86, $P = 3.9e-12$) and the number of cells in each cluster (Pearson's correlation coefficient 0.86, $P = 1.6e-12$) (Fig. 4c; Fig.S8b). These results indicate the significant similar effect on

gene expression of neurons in hypothalamus with P57 and PL treatment.

We further statistically analyzed the overlapped DEGs between P57 and PL treatment in each cluster (Fig. 4d). The KEGG functional enrichment analysis on the intersection genes in cluster 12 and 13, the top two clusters in terms of ratio of overlapped DEGs between P57 and PL treatment, shows that GABAergic synapse and glutamatergic synapse were significantly enriched (Fig. S8c, d). The third top cluster 6 is anatomically located in MPA, the subregion of POA of the hypothalamus, according the Harmonizome database[48]. We selected the top 6 marker genes for MPA (Pbx1, Esr1, Uba1, Ndfip1, Tex22, Pgr), and found that these marker genes are highly expressed in clusters 4,6,7 and 10, suggesting that these clusters likely to contain a significant number of neurons from the MPA (Fig. S8e). Zhang et al reported that activating ERα⁺ MPA neurons was sufficient to drive a torpor-like state in rodents[35]. For these clusters, we performed enrichment analysis for DEGs between P57/PL and control groups. Results shows that there was significant overlap in gene expression between P57 and PL in the MPA (Fig. S8f), and KEGG pathway analysis suggests both small molecules upregulate glutamatergic synapse and downregulate GABAergic synapse signaling pathways (Fig. S8g).

### POA is the potential target region for P57 and PL to induce hypothermia

To investigate the effect of P57 and PL on the activities of MPA neurons during regulation of the core temperature, we monitored the expression of immediate early gene – c-Fos upon intraperitoneal administration of either P57 or PL. We observed three regions of marked activation of c-Fos in mouse brain, that is MPA (Fig.S9a, b), bed nucleus of the stria terminalis, lateral division, dorsal part (BSTLD) and paraventricular nucleus of hypothalamus (PVH), but not in the lateral preoptic area (LPO) or vLPO (Fig. S9–f). We directly injected low doses of P57 (150 μg/kg) into MPA, BSTLD, PVH or LPO, only the MPA group mice induced hypothermia phenotype (Fig.S9g–j), which suggested that MPA may play an important role in P57-induced hypothermia. We then directly recorded the activity of MPA neurons in slice with patch-clamp technique. Either P57 or PL application tended to increase the firing rate of MPA neurons (Fig. S10a–e). Finally, with in vivo fiber photometry calcium signal recording, we simultaneously monitored the changes of MPA GABA or glutamate neuronal activity based on GCaMP (Fig. S10f) and mouse core body temperature. GCaMP fluorescence signal from both GABA and glutamate neurons tended to increase with the falling of body temperature after intraperitoneal administration of P57 or PL (Fig. S10g–n). In addition, the fluctuation degree of GCaMP also tended to increase with the reduction of core body temperature (Fig. S10g, h and Fig. S10k–n). Together with previous studies showing that increasing the activity of MPA GABA and glutamate neurons could reduce core body temperature[33,37,49], our results suggest that MPA is a possible target of P57 for inducing hypothermia.

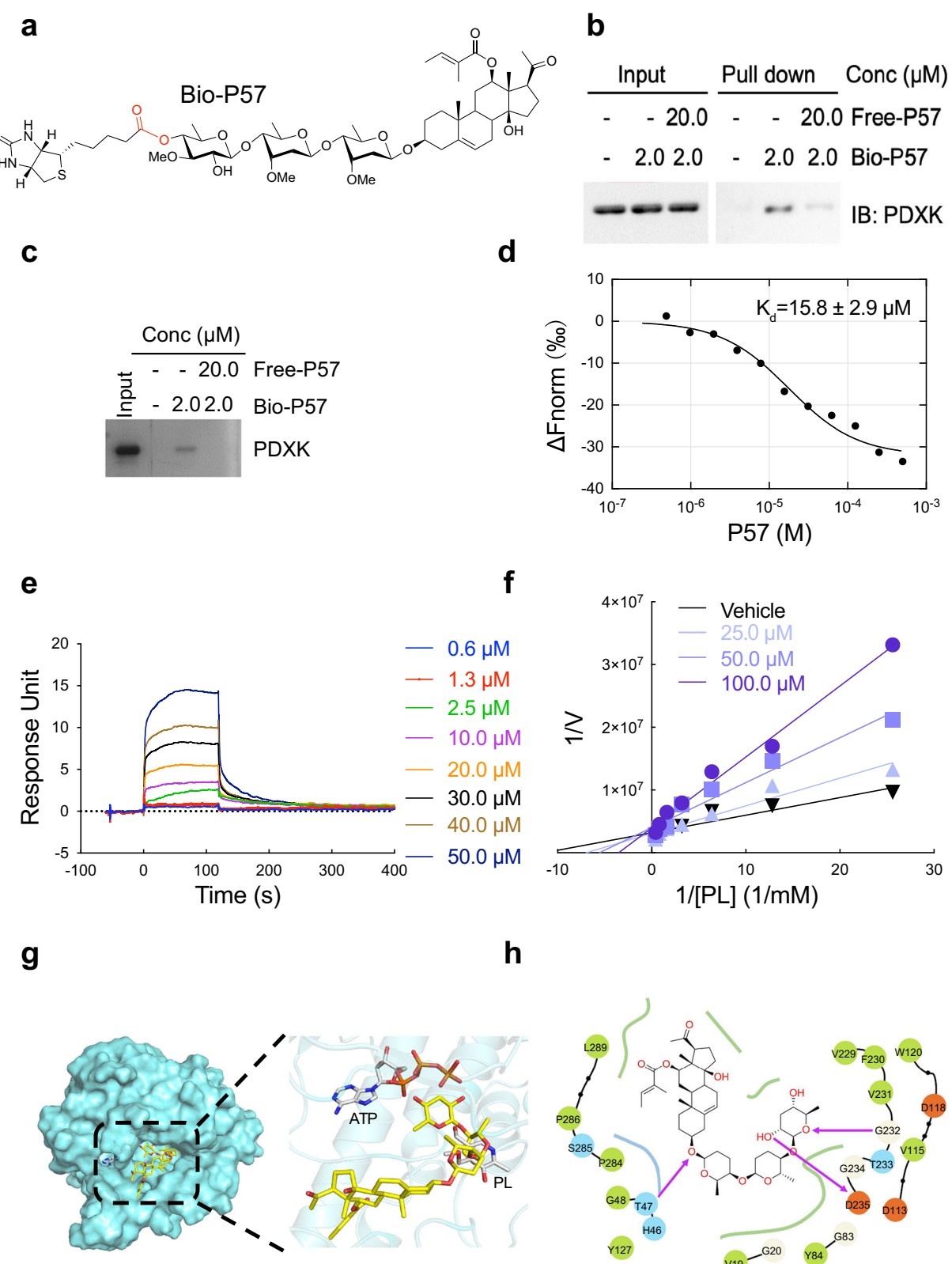

## Discussion

In this study, we demonstrate that P57 robustly induces hypothermia in mice through targeting PDXK, the key enzyme of vitamin B6 metabolism (Fig. 5). P57 administration rapidly decreases core body temperature within minutes and the hypothermic state can be maintained by adjusting the drug dose. We further investigate the possible targets of P57 in central nervous system, and find that POA in hypothalamus is

a key candidate region. Previous studies have shown that activation of MPA neurons in POA can reduce body temperature[35,36]. Several evidence from our study suggest that MPA may be the potential target of P57 to induce hypothermia, including knocking down PDXK in POA reduced the P57-induced hypothermia, P57-induced activation of MPA neurons, local microinjection of P57 into MPA but not into BSTLD, PVH or LPO, induced the hypothermia. However, micro-injection of 250 nL

**Fig. 2 | P57 binds with PDXK and inhibits its enzyme activity. a** Chemical structure of biotinylated P57 probe. **b, c** Affinity chromatography experiment with mouse brain lysate **b** or purified protein PDXK **c** using P57-Bio (2.0 μM) probe in the absence (middle lane) and presence (right lane) of P57 (20.0 μM). **b** Western blot was performed to measure the binding amount of PDXK with P57-Bio. **c** Silver staining was performed to measure the binding amount of PDXK with P57-Bio. **d** Dose–response curve for the binding interaction between P57 and PDXK using MST. The concentration of purified protein PDXK is kept constant, while the concentration of P57 varies from 0 to 100 μM. The binding curve yields a dissociation constant $K_d = 15.8 \pm 2.9$ μM. **e** A surface plasmon resonance (SPR) assay characterizing the binding between P57 and purified protein PDXK. Color lines, model fits of

SPR data from different concentrations of P57. The calculated dissociation constant $K_d = 3.5$ μM. **f** Kinetic analysis of PDXK inhibition by P57 with respect to the substrate pyridoxal (PL) shows that P57 acts as a competitive inhibitor. **g** A surface rendering of predicted binding mode of P57 (stick rendering) at the active site of human PDXK (cyan). Interactions between P57 and surrounding residues of PDXK, critical residues are rendered in stick representation (C, cyan; N, blue; O, red; H, gray), hydrogen-bond interactions are shown as black dashed lines. **h** 2D-dimensional interaction schemes of predicted binding pose of P57 in ATP/PL active site generated by Maestro[47]. Magenta lines with arrow denote hydrogen bonds. Source data are provided as a Source Data file.

P57 may spread to other areas than MPA, and the contribution of MPA tissue lesions to P57-induced hypothermia cannot be ruled out. In addition, knocking down PDXK in POA with AAV viruses did not completely block the P57-induced hypothermia, we speculate that this may due to the efficiency of virus infection, the contribution of the peripheral effects, and the possibility of P57 targeting other brain regions to induce hypothermia. We observed that P57 treatment reduced BAT and tail temperatures in mice (Fig.S1b, c), and there are reports that the DMH contains BAT sympathoexcitatory neurons that are synaptically-connected to BAT[50], the MPA contains GABAergic neurons projecting to the DMH neurons[51], these also suggests that P57 may affect BAT thermogenesis through the projection of MPA to DMH. In addition, we also find that P57 has remarkable neuroprotective effects in both mouse and rat MCAO models (Fig. 1h-j, and Fig.S3), indicating that P57 may act as a potent new therapeutic approach to treat ischemic diseases.

Target identification of natural products is one of the main bottlenecks in drug development. Previous studies showed that P57 significantly suppressed appetite activity and decreased body weight[40,52,53], but the underlying mechanisms are unclear, which is the one of limiting factors for its clinical application[45,52,53]. Identification of PDXK as the target of P57 (Fig. 2) provides insight into its mechanism of actions and will prompt structural optimization of P57 to be applied in clinical settings. Interestingly, P57 inhibits PDXK enzyme activity to specifically accumulate PL in hypothalamus (Figs. 2f and 3a), resulting in a rapid decrease in core temperature and basal metabolic rate. We do not exclude the possibility that mechanism of thermoregulation by P57 and PL is related to PLP, which is a cofactor for many metabolic enzymes, though it is unlikely because PL administration is sufficient to induce hypothermia (Fig. 3h–j, Fig. S6f). Defects of vitamin B6 metabolism in the brain leads to early onset of vitamin B6-dependent epilepsy refractory to anticonvulsants[54]. *PDXK* mutation leads to peripheral neuropathy with optic atrophy[55]. The mechanisms involved in these neurological diseases are largely unknown. P57 will be a useful tool to investigate PDXK and vitamin B6's function in these neuronal pathological processes.

Hypothalamic POA serves as an essential brain region in maintaining and regulating body temperature[28–31]. As a multifunctional region, POA can be divided into several nuclei and is composed of different types of neurons to form complex neural circuits and modulate body temperature, sleep, feeding, sexual behavior, and so on[28,56]. In our study, we identify that POA may be the key brain target region of P57 to induce hypothermia. Several neuronal subpopulations in POA are uncovered to directly regulate core body temperature, including glutamatergic LepRb neurons[57], *Trpm2*+/*Vglut2*+ POA neurons[58], VMPO[BDNF/PACAP] neurons[36], GABAergic neurons in the vLPO[33], ERα+ neurons in the MPA[35], Q neurons (encoding pyroglutamylated RF amide peptide, QRFP) within AVPe/MPA[37] and *Vglut2*+*Adcyap1*+ neurons[49]. We found P57 and PL significantly increased c-Fos expression in MPA, BSTLD and PVH, but microinjection of P57 into the MPA is able to induce hypothermia similar as the effect of intraperitoneal injection. According to our single-cell sequencing results, electrophysiological recording, and fiber photometry recording, we further find that P57 and PL can regulate both glutamatergic and GABAergic

neurons in MPA, which suggest that P57 may excite these neurons to induce hypothermia. Further study combing TRAP mouse line[59] and optogenetic/chemogenetic tools may help us dissect the specific MPA neuron and neural circuit involved in the P57-induced hypothermia. In addition to thermoregulation, POA is also involved in regulating sleep and feeding behavior[28,56,60,61]. P57 is well-known for its prominent activity in appetite suppression with undefined mechanism[39,40], so P57 may also target POA to regulate feeding behavior and sleep. There are several questions left to be further investigated, including: (1) the mechanism(s) of how P57 and PL activate these neurons, directly via inhibition of PDXK or indirectly via synaptic transmission; (2) whether MPA glutamatergic and GABAergic neurons are both involved on the P57-induced hypothermia; (3) the neuronal circuit(s) participated in hypothermia induced by P57. Nonetheless, our study suggests that MPA may be an important region of P57 for inducing hypothermia via inhibiting PDXK and P57 will be a great probe for exploring the above questions in the future for the thermoregulation and others.

Our results showed that P57 reduces the infarct volume in a MCAO model of stroke, and maintaining mouse temperature abolishes the neuroprotective role of P57. Though there are differences in body temperature regulation between mice and rat[62,63], we observed similar neuroprotective effects of P57 in these two species. Thus, potent effects of P57 and PL demonstrate feasibility of pharmacological induction of therapeutic hypothermia state in mammals. Furthermore, identification of PDXK as a target of P57 reveals previously unknown roles of vitamin B6 metabolism in thermoregulation and will offer an opportunity for future clinical drug development to control hypothermia and other related disease processes.

## Methods
### Data reporting
No statistical methods were used to predetermine sample size. Animals used were randomly assigned to groups before experiments and the investigators were not blinded to allocation during experiments and outcome assessment.

### Animals
All animal care and experimentation were ethically performed according to procedures approved by the Institutional Animal Care and Use Committee at Fudan University. Mice were housed 3–5 per cage (except in metabolic chamber in which mice were singly caged or preparing for core temperature detection) in a 12 h light/12 h dark cycle with ad libitum access to regular chow (P1101F, slacom) and water, and maintained at an ambient temperature of 23 °C and a relative humidity of 50%. We used 6–8-week-old WT male C57BL/6 J mice (SLAC company), Wistar rats (SLAC company), Vglut2-ires-Cre (Jackson Lab), Vgat-ires-Cre (Jackson Lab). All mouse lines are in a WT (C57BL/6 J) background.

*Pdxk*[flox/flox] mice were generated by CRISPR/Cas 9-mediated genome engineering. Exon2 of *Pdxk-201* (ENSMUST00000041616.14) transcript, containing 55 bp coding sequence, is recommended as the knockout region based on the structure of *pdxk*. In brief, sgRNA was transcribed in vitro, donor vector was constructed. Cas9, sgRNA and donor were microinjected into the zygotes of C57BL/6JGpt mice.

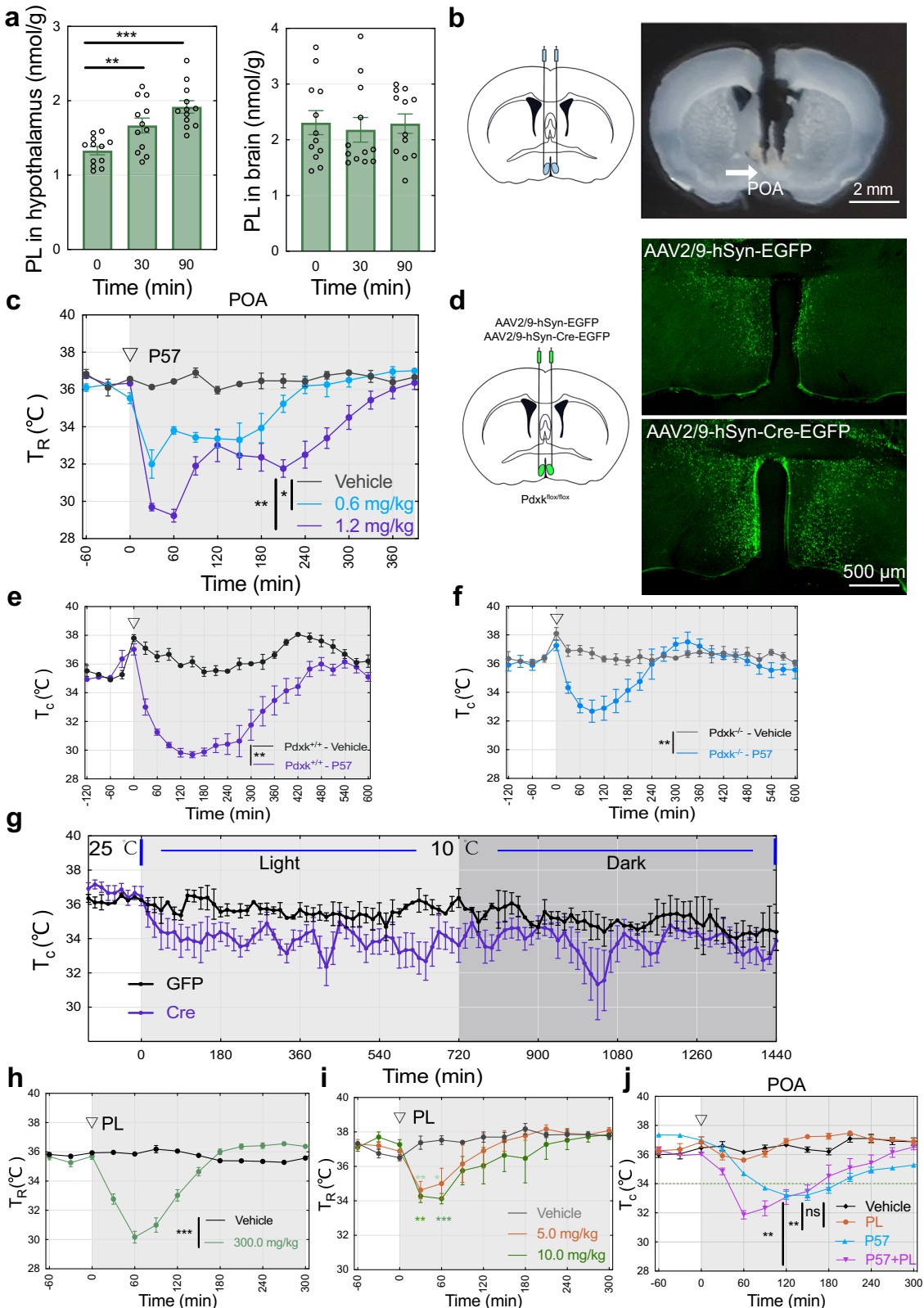

Zygotes were transplanted to obtain positive F0 mice which were confirmed by PCR and sequencing. A stable F1 generation mouse model was obtained by mating positive F0 generation mice with C57BL/6JGpt mice.

### Chemicals

P57[64] and its derivatives were kindly provided from Biao Yu's group by totally synthesized. Pyridoxal (PL, 322481) and pyridoxal 5'-phosphate (PLP, 303712) were obtained from J&K Scientific. Pyridoxamine (PM, P9380), pyridoxine (PN, P9755), 4-pyridoxic acid (4-PA, P9630), pyridoxal-(methyl-d3) (D3-PL, 705187), celastrol (cela, C0869), ethacrynic acid (EA, SML1083) and 2,3,5-triphe-nyltetrazolium chloride (TTC, 8877) were obtained from Sigma Aldrich. α-angelica lactone (HY-N0548) was obtained from MCE. Glutathione (GSH, 101814) was obtained from MP Biomedicals.

**Fig. 3 | P57 mainly targets PDXK in hypothalamus to induces hypothermia.**
**a** LC-MS detection of PL in hypothalamus (left) and brain tissues (right). Hypothalamus and brain tissue were sampled at 0 min, 30 min and 90 min after treated with P57 (25.0 mg/kg), $n = 12$ mice, student's t test. **b** Schematic showing the bilateral injection of drug into the POA. Coronal brain sections showing location of cannula implanted in POA. Scale bars, 2 mm. **c** Rectal temperature of mice treated with P57 in the POA. 15 mg/mL or 30 mg/mL P57 (0.6 mg/kg or 1.2 mg/kg,) was injected into POA bilaterally at 0 min (arrow) (400 nL for each laterally). $n = 3$ mice, two-way ANOVA. **d** Schematic showing the bilateral injection of virus into the POA. Coronal brain sections showing EGFP (upper right) or Cre-EGFP (lower right) expressed in POA. Scale bars, 500 μm. **e**, **f** Core temperature of P57-treated mice with conditional knockout PDXK or not in POA. Virus were injected into hypothalamus bilaterally and let the mice recover for one month, then P57 (25 mg/kg) was injected intraperitoneally, $n = 3$ mice, two-way ANOVA. **g** Core temperature of mice with conditional knockout PDXK or not in POA. Mice were treated with cold exposure (10 °C) for 24 h, $n = 3$ mice. **h** Core temperature of PL-treated mice. PL was injected intraperitoneally, $n = 6$ mice, two-way ANOVA. **i** Core temperature of mice treated with PL in the POA. 100 mg/mL or 200 mg/mL PL (5.0 mg/kg or 10.0 mg/kg) was injected into POA bilaterally (500 nL for each laterally), $n = 4$ mice, student's t test. **j** Core temperature of mice treated with combination of PL (200.0 mg/kg) and P57 (12.5 mg/kg). PL or P57 was injected intraperitoneally into C57BL/6 J male mice, $n = 5$ mice, two-way ANOVA. Student's t test used was two-sided. All error bars are presented as mean values ± s.e.m. * $P < 0.05$, ** $P < 0.01$, *** $P < 0.001$ and ns, not significant. Source data are provided as a Source Data file.

## Drug administration

P57 solution was prepared by initially dissolving 10 mg P57 in 40 μL DMSO, and then add 3960 μL 10% castor oil (1:9 mixture of castor oil: PBS) for intraperitoneal injection. Vehicle control means a mixture of 1% DMSO, 10% castor oil and 89% PBS. PL solution was prepared by dissolving 30 mg PL in 1 mL PBS for intraperitoneal injection. Vehicle control of PL is PBS.

For local injection into subregion of brain, cannula was pre-implanted in interested location. The cannula coordinates of subregion of brain were calculated according to Paxinos & Franklin mice brain coordinates (3rd edition). For Fig. 3c, i, the cannula was implanted in the POA (anterior-posterior (AP), 0.22 mm; medial- lateral (ML), ±0.25 mm; dorsal-ventral (DV), −5.20 mm). For Fig. S9g, the cannula was implanted in the MPA unilaterally (AP, 0.22 mm; ML, −0.25 mm; DV, −5.20 mm). For Fig. S6b, the cannula was implanted in striatum (AP, 0.22 mm; ML, 2.00 mm; DV, −2.35 mm). For Fig.S9h, the cannulas were implanted in the BSTLD (AP, 0.22 mm; ML, 0.90 mm; DV, −4.00 mm). For Fig. S9i, the cannulas were implanted in the PVH (AP, −0.80 mm; ML, 0.20 mm; DV, −4.50 mm). For Fig. S9j, the cannulas were implanted in the LPO (AP, 0.22 mm; ML, −0.75 mm; DV, −5.20 mm). Animals were anaesthetized by 3% isoflurane and hypothalamic injection cannula were fixed with dental cement to three stainless-steel screws inserted into the skull. Ketoprofen was injected intraperitoneally to reduce pain during and after surgery in mice. Mice were recovered in a warm bracket and then transferred to housing cages. After two weeks, mice were implanted with Anilogger® core temperature monitoring system into their abdominal cavity. The temperatures were record with hypothalamic injection of compounds after another one week to wait for the mice to fully recover.

For local injection into subregion of brain, P57 solution was prepared by dissolving 15 mg or 30 mg P57 in 10 μL DMSO, and then add 990 μL 10% castor oil. Vehicle control is a mixture of 1% DMSO, 10% castor oil and 89% PBS. PL solution was prepared by dissolving 200 mg PL in 1 mL PBS. Drug was delivered through cannula at 4 nL/s using a Hamilton syringe and syringe stayed for another 3 min. For Fig. 3c, 0.8 μL of 15 mg/mL or 30 mg/mL P57 was injected into POA bilaterally (400 nL for each laterally). For Fig. 3i, 1 μL of 100 mg/mL or 200 mg/mL PL was injected into POA bilaterally (500 nL for each laterally). For Fig. S9g-j, 0.2 μL of 15 mg/mL P57 was injected into MPA, BSTLD, PVH or LPO unilaterally.

For the brain slice patch clamp, P57 was dissolved in DMSO at 20 mM. The working concentration of P57 is 2 μM (1:1000 diluted with artificial cerebrospinal fluid (ACSF)).

## Viruses

AAV2/9-hSyn-DIO-jGCaMP7s (S0590-9), AAV2/9-hSyn-eGFP-WPRE-pA (S0237-9-H20), and AAV2/9-hSyn-Cre-eGFP-WPRE-pA (S0230-9-H50) were purchased from TaiTool. All virus were diluted with PBS to a final concentration of $5 \times 10^{12}$ per mL before stereotaxic delivery into the mouse brain.

For injection of AAV2/9-hSyn-DIO-jGCaMP7s into MPA, Vglut-Cre and Vgat-Cre mice (6–8 weeks old) were anaesthetized with isoflurane and positioned in a stereotaxic frame (David Kopf Instruments). 180 nL (90 nL per site) of virus was delivered into MPA (AP, 0.22 mm; ML, −0.25 mm; DV, −5.4/−5.20 mm) at a controlled rate of 2 nL per second using a Hamilton needle syringe. The needle was kept in place for 10 min after injection. The waiting period for recovery and virus expression for the experiments was 3–4 weeks.

For injection of AAV2/9-hSyn-eGFP-WPRE-pA and AAV2/9-hSyn-Cre-eGFP-WPRE-pA, $Pdxk$^flox/flox^ mice (6–8 weeks old) were anaesthetized with isoflurane and positioned in a stereotaxic frame. 100 nL of virus per site was delivered into POA (AP, 0.22 mm; ML, ± 0.25 mm; DV, −5.4/−5.20 mm) at a controlled rate of 2 nL per second using a Hamilton needle syringe.

## Body temperature recording

The rectal temperatures were recorded by animal temperature measuring apparatus (FT3400, Kew Basis) (Figs. 1i; 3c, h, i; Fig.S3a; Fig.S4c–e and Fig. S6f) every 30 min and the surface temperature was recorded by IR digital thermographic camera (FLIR T430sc) every 10 minutes (Fig.S1b, c). The other internal temperatures were measured and recorded by Anilogger® core temperature monitoring system every 15 minutes. The mice were adapted for 4 days before test. Figure3g was exposed to an ambient temperature of 10 °C. All other temperature recording experiments were performed under ambient temperature at 22 - 24 °C.

## Middle cerebral artery occlusion (MCAO) and infarct volume evaluation

Adult male wistar rats were anesthetized with isoflurane. Rats were subjected to a right side MCAO[12,65] and heating lamps were utilized to maintain rectal temperature at 36.5 to 37.5 °C. Reperfusion was achieved by the withdrawal of the filament after 2 h occlusion. In all ischemia models, solvent or P57 was immediately injected intraperitoneally once priming reperfusion.

Infarct volume was evaluated by TTC staining[12]. Briefly, rats were sacrificed at 24 h following reperfusion and their brain were quickly removed and chilled. Six coronal brain slices with a 2-mm thickness were cut for the treatment with 1% TTC solution at 37 °C for 20 min, then fixed in 10% formalin solution. The images of the coronal slices were taken with a digital camera and analyzed to quantify the infarct area with Adobe Photoshop software. Infarct volumes were determined by the integration of the infarct area of each slice and the distances between them.

Mouse MCAO model was done in a similar way with additional group (MCAO + P57 + Warming), which maintained core temperature of mice with heating pad after treatment with P57.

## HE staining

For histopathology analysis, brain, heart, liver, kidney and muscle tissues were fixed by paraformaldehyde and then were dehydrated, embedded in paraffin and cut into serial sections at 5 μm. Sections were stained with hematoxylin and eosin (H&E) solution and observed them under an optical microscope (DP73, OLUMPUS, Japan). The cell

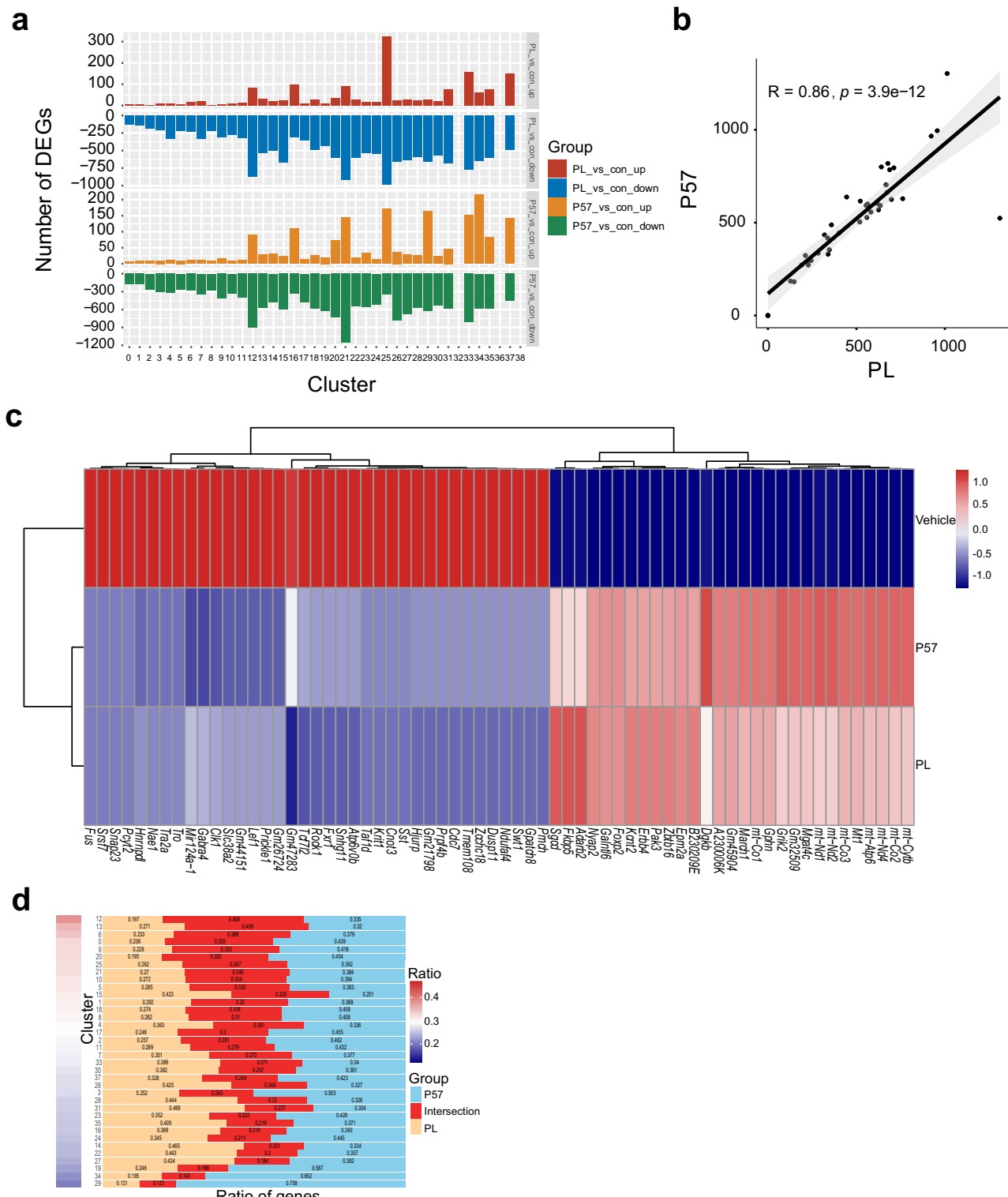

**Fig. 4 | P57 and PL play a similar effect on hypothalamus neurons. a** Bar plot showing the number of DEGs affected by P57 and PL across each cluster in hypothalamic neuron subtypes. **b** Scatter plot shows the correlation of the number of DEGs affected by P57 and PL. Each point represents a cluster, the value means the number of DEGs in each cluster. The linear best fit line is shown, with 95% confidence intervals (shaded areas) and the Pearson correlation coefficient (R) and p-value (P) were calculated (two-sided test). **c** Heatmap of shared DEGs in hypothalamus with P57 and PL treatment. **d** The relative ratio of unique and overlapped DEGs between P57 and PL pair in each cluster. The burlywood and skyblue represents the number of specific DEGs of PL and P57 respectively, and the red stands for the number of overlapped DEGs. The heatmap on the left ranks each cluster by the Jaccard Similarity in DEGs between P57/PL and control group.

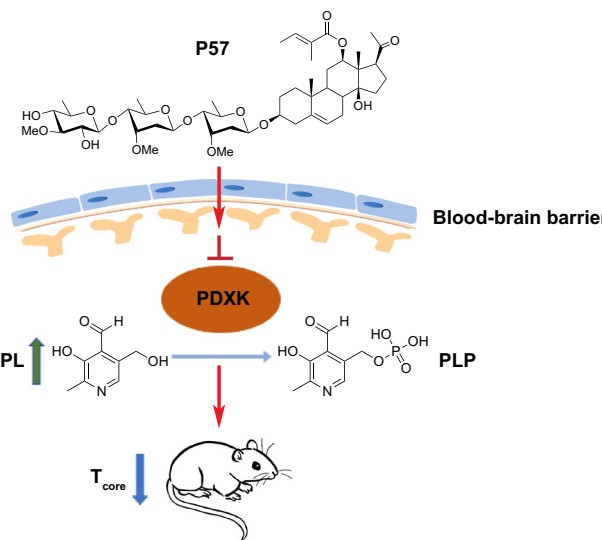

**Fig. 5 | The schematic illustration of P57 regulating body temperature.** The natural product P57 crossed the blood-brain barrier and entered the brain, inhibiting PDXK enzyme activity, leading to the accumulation of PL, and eventually causing the mouse core body temperature to drop.

structure of each tissue in P57 treatment or vehicle was complete and normal with clear boundaries. No inflammatory cell aggregation, hyperplasia, chromatin concentration, nucleus fragmentation and other abnormal pathological changes were observed.

### Affinity chromatography experiment and LC-MS/MS analysis

A total of 500 μL brain lysate was pre-incubated 1 h with DMSO or 20 μM P57 at 4 °C with rotation, and then incubated 1 h with DMSO or 2 μM P57-Bio at 4 °C with rotation. The samples were added to 20 μL pre-washed high-capacity streptavidin agarose beads and incubated 1 h at 4 °C with rotation. The beads were pelleted and washed 4 times with lysis buffer. The wash buffer was aspirated completely from the beads. Then 50 μL 1× sample buffer was added and incubated for 5 min on a 95 °C-heat block. The sample buffer containing proteins off the beads was carefully transferred to a new 1.5 mL Eppendorf tube, and was loaded onto 15% SDS-PAGE.

### In-gel digestion

After silver staining, the gel was washed with deionized water and gel lanes were excised using surgical scissors into pieces. Then gel pieces were reduced with 5 mM dithiothreitol (DTT) at 56 °C for 1 h. Alkylation was followed by incubation with 15 mM iodoacetamide (IAA) in dark for 45 min. After removed the supernatant, the samples were incubated with trypsin covering at 37 °C overnight. Transferred the supernatant into a clean tube. Squeezing the gels used 100% ACN (v/v) to ensure all the peptides could be extracted. Combined all the peptides and dry them by Speed-Vac. Then the peptides were dissolved in 0.1% TFA and further desalted with C18 Zip-tip.

### LC−MS/MS analysis

LC−MS/MS analysis was conducted using an EASY-nLC 1000 HPLC system (Thermo Fisher Scientific) and an Orbitrap Fusion (Thermo Fisher Scientific). The sample was reconstituted using buffer A (0.1% formic acid, 2% ACN) and then loaded onto C18 reversed phase capillary analytical column (3 μm particle size, 90 Å pore size) by the autosampler which connected to an EASY-nLC 1000 HPLC system. The peptides were eluted at constant flow rate of 300 nL/min by increasing buffer B (0.1% formic acid, 90% ACN) from 8% to 32% over 58 min, then 48% in 6 min followed a steep increase to 80% in 2 min.

The mass spectrometric analysis was carried out in a data dependent acquisition (DDA) mode with a cycle time of 3 s. The peptides with a range of m/z 350−1300 were detected by an Orbitrap mass analyzer. And the resolution was set as 120,000 at m/z 200. Automatic gain control (AGC) target and maximum ion injection time (IT) were $5.0 \times 10^5$ and 50 ms, respectively. The isolated precursor ions were subjected to fragmentation via higher-energy collisional dissociation (HCD) with a collision energy of 32%, and analyzed by ion trap analyzer. The dynamic exclusion was set as 60 s, and the charge inclusion state was set to $2-6+$.

### Data Analysis

The raw data were analyzed by Mascot search engine (v2.3.0) against the UniProt Mus musculus database (downloaded 2013). The parameters were set as following: Carbamidomethylation (C) and acetylation (protein N-terminal) were set as fixed modifications and oxidation (M) as variable modification. Trypsin was specific enzyme and allowed up to 2 missed cleavage. Mass tolerances were set to ±10 ppm for precursor ions, ±0.5 Da for fragment ions. Fiji's Transfom-Landmark Correspondences and Trainable Weka Segmentation plug-in were used for brain sections registration and c-Fos signal statistics respectively.

### Western immunoblotting

Whole brain protein lysate was extracted using buffer containing 20 mM Tris-HCl (PH = 7.4), 100 mM KCl, 1% Triton-X-100, and complete protease inhibitors. The following antibodies were used for immunoblotting: PDXK (1:1000; 15309, Proteintech), GSTP1 (1:1000, 15902, Proteintech) and GAP43 (1:1000, 16971, Proteintech). Membranes were incubated overnight with primary antibody at 4 °C, followed by 3X washes in PBST (PBS with 0.1% Triton-X-100, vol/vol) before incubation in secondary antibodies Peroxidase AffiniPure Goat Anti-Rabbit IgG (H + L) (1:10000, 111-035-003, Jackson ImmunoResearch Inc) for 1 h at room temperature. Images were captured using ChemiScope system (Clinx Science Instruments Co. Ltd).

### Protein expression and purification

The coding sequences of wild-type full-length human PDXK and the site-directed mutant proteins (T47V, Y84E, D235A) were respectively subcloned into pET-28a vector resulting in the addition of an N-terminal His tag and thrombin cleavage site. The plasmids were transformed into Escherichia coli OverExpress C43(DE3) strain (2nd Lab, cat# EC1040) and induced with 0.4 mM IPTG at 18 °C overnight. The fusion proteins were purified by HisTrap FF column (GE Healthcare Life Sciences) after lysis of cells by sonication, followed by size-exclusion chromatography using a Superdex 75 column (GE Healthcare Life Sciences). Once purified, the proteins were subsequently frozen in aliquots and stored in buffer (25 mM HEPES-NaOH, pH 7.5, 200 mM NaCl, and 5% glycerol).

### Microscale thermophoresis (MST) assay

A Monolith NT. Automated from NanoTemper Technologies was used for MST binding assay. Wild-type or mutant PDXK proteins were fluorescently labeled with the RED-tris-NTA (NanoTemper, cat# MO-L008) according to the manufacturer's instructions. All affinity measurements were performed in PBS buffer mixed with 0.05% Pluronic F-127 (Invitrogen, cat# P6866). Indicated compounds, arrayed at different concentrations, were incubated with proteins for 30 min before applied to Monolith NT standard treated capillaries. Thermophoresis was then determined at 25 °C with 15−20% excitation power and middle MST power. The data analysis was performed using the analysis software (MO. Affinity Analysis) after the measurement and plotted by GraphPad Prism 8.

## Surface plasmon resonance (SPR) assay

SPR experiments were performed on Biacore T200 instrument (GE Healthcare). PDXK protein was covalently immobilized onto CM5 chip as standard procedure. The chip was equilibrated overnight first with HBS buffer (10 mM HEPES (PH = 7.4), 150 mM NaCl, 0.05% (v/v) surfactant P20 and 0.2% dimethyl sulfoxide). P57 was gradiently diluted with HBS buffer and then injected at 30 μL/min for 120 s contact time followed by dissociation for 300 s. Sensorgrams were fitted to the Langmuir binding equation for a 1:1 interaction model using Biacore T200 Evaluation Software v3.0 to determine kinetic parameters and equilibrium dissociation constants. Each experiment was performed at least three times.

## Determination of kinetic constants

All PDXK kinetic measurements were carried out at 37 °C with 125 nM protein in 384-well plates (Corning, cat# 3702). Initial velocity studies for the conversion of PL to PLP were performed at 380 nm in an Agilant 8454 spectrophotometer (Thermo Fisher) in 100 mM sodium HEPES buffer, pH 7.4. The concentration of PL (J&K, cat# 322481) and Mg-ATP (Sigma, cat# A2383) were fixed at 2 mM and 1 mM respectively when titrating another substrate. Values for Km and $K_{cat}$ were determined from Michaelis-Menten by GraphPad Prism 8.

## Molecular docking

Docking studies were performed with methods as previously published (https://doi.org/10.1016/j.ejmech.2019.111767) using Schrödinger suite (version 2009), which includes all of the programs described below. P57 was prepared through LigPrep panel integrated to generate all low-energy stereoisomers and possible ionization states in the pH range of 7.0 ± 2.0 with Epik. The crystal structure of PDXK in complex with PLP and ATP (PDB code 3KEU) was prepared using the Protein Preparation Wizard with default parameters. The ligand and protein were energy-minimized using OPLS-2005 force field. A PLP-centered receptor grid was generated with the program Glide. Then docking study was performed using Glide in extra-precision (XP) mode with enhanced planarity of conjugated pi groups, and strain correction terms applied.

## Rotarod test

For the rotarod test, the mice were trained 2 days before test. On day 3, mice were placed on an accelerating rotarod cylinder, and the latency time of the animals was measured. The speed was increased from 5 to 40 rpm within 240 s. A trial ended if the animal fell off the rungs or gripped the device and spun around for 2 consecutive revolutions without attempting to walk on the rungs. Motor test data are presented as the mean of latency time on the rotarod.

## Open-field test

On the first day, the baseline of motor activity was recorded before treatment. Each mouse was gently placed in the center of the box and the motor activities were recorded for 20 min. Then the motor activity was recorded at 30 min after intraperitoneal injection of Vehicle, P57 or PL. On the second day, the motor activity was also recorded when the core temperature of mice recovered to normal level. Total distance traveled was recorded using the Ethovision XT video tracking software system (Noldus Information Technologies, Leesburg, VA, USA).

## Metabolic studies (CLAMS)

Around 6–8-week-old C5BL/6 J male mice were maintained individually in a metabolism chamber (Comprehensive Lab Animal Monitoring System, CLAMS) with free access to food and water for 72 h. Mice were housed for 24 h for adaption. On day 2, mice were intraperitoneally injected with solvent or P57 (12.5 mg/kg or 25 mg/kg). Metabolic parameters including $O_2$ consumption, $CO_2$ production, respiratory exchange ratio (RER), energy expenditure and total locomotor activity

were recorded at 24 min intervals in a standard light-dark cycle (light 7:00–19:00 and dark 19:00–7:00) at 25 °C. The respiratory quotient is the ratio of carbon dioxide production to oxygen consumption ($VO_2$). $RER = VCO_2/VO_2$. Energy expenditure was calculated as the product of the calorific value of oxygen ($3.815*VO_2 + 1.232*VCO_2$)[66].

## Measurement of PL level in hypothalamus

Pyridoxine (PN), pyridoxamine (PM), pyridoxal (PL), 4-pyridoxic acid (4-PA), and pyridoxal 5′-phosphate (PLP) were analyzed on an Agilent 1260 Infinity liquid chromatography (Agilent, CA, USA) coupled with an AB SCIEX QTrap 6500 Mass Spectrometer (AB SCIEX, MA, USA) according to a method reported by Midttun et al. previously[67]. Pyridoxal-(methyl-d3) (PL-d3) was used as the internal standard to tract the extraction efficacy. The accuracy of each analyte was validated to be of more than 90% with the RSD of less than 10%.

## Acute slice preparation and electrophysiology

Mouse brain slices were prepared according to previously published procedures[68]. Mouse was deeply anesthetized by isoflurane, and followed by transcardial perfusion with ice-cold, oxygenated (95% $O_2$/5% $CO_2$) artificial cerebrospinal fluid (ACSF) containing (in mM): 127 NaCl, 2.5 KCl, 25 NaHCO₃, 1.25 NaH₂PO₄, 2.0 CaCl₂, 1.0 MgCl₂, and 25 Glucose (osmolarity ~310 mOsm/L). After perfusion and decapitation, mouse brain was removed and immersed in ice-cold and oxygenated ACSF. Tissue was blocked and supported by a block of 4% agar, and transferred to a slicing chamber containing ice-cold and oxygenated ACSF. Coronal slices of 250-μm-thick were prepared on a vibratome (Series1000, Tissue Sectioning System, Natural Genetic Ltd., USA), in rostral/caudal direction. Slices were transferred to a chamber containing constantly oxygenated ACSF, and incubated at around 34 °C for 20 min before recording.

Brain slices were transferred to a recording chamber perfused with oxygenated ACSF at a flow rate of 1.5-2 ml/min, and temperature was maintained at ~ 30 °C during recording by feedback in-line heater (TC-324C; Warner Instruments, Hamden, CT). MPA GABA and non-GABA neurons were visualized in slices using IR/DIC microscopy, and identified based on tdTomato signal in Vgat-ires-Cre; Ai14 mice. Cell-attached recordings were established with glass pipettes (3–5 MΩ) containing the following (in mM): 135 K-gluconate, 4 KCl, 10 HEPES, 10 Na-phosphocreatine, 4 MgATP, 0.4 Na₂GTP, and 1 EGTA (with pH 7.2–7.3, and osmolarity ~295 mOsm/L). P57 (2 μM) or PL (10 μM) were applied by bath perfusion. Recordings were made using 700B amplifier and Digidata 1440 A interface (Axon Instruments, Union City, CA). Signals were filtered at 4 kHz, sampled at 10 kHz, and analyzed using Clampex 10.7 (Molecular Devices).

## Tissue processing, immunohistochemistry and imaging

Mice were deeply anesthetized with isoflurane, and perfused transcardially with 4% paraformaldehyde (PFA) in 0.1 M phosphate buffered saline (PBS). Brains were removed and post-fixed for 24 hours in 4% PFA at 4 °C, followed by dehydrating with 20% and 30% sucrose for 24 h in sequence. Dehydrated brains were embedded in OCT, and cut at 40 μm using a cryostat (CM1950, Leica). Tissues were chosen and pretreated in 0.2% Triton-X100 for an hour at room temperature (RT), then blocked with 0.05% Triton-X100, 10% bovine serum albumin (BSA) in PBS for one hour at RT and rinsed in PBS. Tissues were transferred into primary antibody solution (Rabbit anti-cFos, 1:1000, 5348 S, Cell Signaling) in PBS with 0.2% Triton-X100 and incubated for 24 hours at 4 °C. Tissues were rinsed in PBS three times, and incubated with secondary antibody solution (Goat anti-Rabbit 647, 1:1000, A21244, Thermofisher) in PBS for 2 hours at RT, then rinsed with PBS for three times and mounted onto slides, dried and covered under glycerol: TBS (3:1) with Hoechst 33342 (1:1000, ThermoFisher Scientific). Sections were imaged with an Olympus VS120 slide scanning microscope. Confocal images were acquired with a Nikon A1 confocal

laser scanning microscope with an X25 objective. Images were analyzed in ImageJ (FIJI).

## In vivo fiber photometry calcium signal recording

For in vivo fiber photometry recording, animals were anaesthetized by 3% isoflurane and positioned in a stereotaxic frame (David Kopf Instruments). Ketoprofen was injected intraperitoneally to reduce pain during and after surgery in mice. AAV-hSyn-DIO-jGCaMP7s viruses were injected into MPA of Vgat-ires-Cre and Vglut2-ires-Cre mice with a microsyringe pump controller (NanoJect III, Drummond Scientific Company). Three weeks after virus injection, an optical fiber (400 μm, 0.57 NA, MFC_400/430, Doric Lens) was implanted into MPA. During optical fiber implantation, we monitored the change of fluorescence intensity in real-time with photometry recording system (Doric Lens). GCamp7s was excited with sinusoidally modulated light from laser diode modules at 465 nm (211 Hz) and 405 nm (531 Hz). Signals emitted from GCamp7s and its isosbestic control channel were acquired using a photoreceiver (H10722-20; Hamamatsu Photonics), digitized at 500 Hz, and then recorded by Doric Neuroscience Studio Software. The final depth of implantation was determined by the cessation of further increases of fluorescence intensity, and optical fiber was secured with dental cement. Mice were housed for at least 1 week for recovery before recording. During recording, mouse was placed in its home cage with an optical fiber patch cord connected to the implanted optical fiber. Changes in fluorescence with time were calculated by smoothing signals from the isosbestic control channel. Relative fluorescence intensity was defined as the signal intensity change relative to the first 5-min baseline recording before i.p. injection. Mouse core body temperature was monitored every 5 min with a thermal camera (FLIR T430sc) during recording. Mice were anaesthetized with ventilated isoflurane at 3%, and then were euthanized by transcardial perfusion of 10 mL saline followed by 10 mL of 4% paraformaldehyde to fix the brain to confirm the virus expression and the location of the recording probe after finishing the experiment. The data was acceptable when the virus was expressed in MPA and the recording probe targeted the MPA.

## Hypothalamic snRNA-seq

Raw data quality (Fastq format) was assessed and poor-quality reads were removed using fastp (version 0.19.5) program[69] with the following parameters: "-n 15 -q 20 -u 50 -w 16". Cleaned data were then mapped to reference genome (mm10) using the default parameters of Cell Ranger v6.0.2 (10× Genomics). Since both un-spliced pre-mRNA and mature mRNA were captured in snRNA-seq strategy, we set the parameter "--include-introns" in Cell Ranger pipeline, which enables us to counts these unspliced reads.

Seurat (version 4.0)[70] was used to process and cluster cells. For this analysis, three S4 objects, one for each group (P57, PL, and control) were created using the *Read10X* and *CreateSeuratObject* functions with the parameter *min.cell = 3* to filter genes expressed in fewer than 3 cells. The percent of ribosomal protein transcripts (percent.RP), the percent of mitochondrial transcripts (percent.MT), the number of genes detected per cell (nFeature_RNA) and the percent of erythrocyte transcripts (percent.HB) were used to assess cell quality. We removed the low-quality cells by setting cut-off: 200<nFeature_RNA < 6000, percent.MT < 10, percent.HB < 5, percent.RP < 3. To eliminate the effects of different sequencing depth, we normalized the data using *NormalizeData* function with the following parameters: normalization.method = "LogNormalize", scale.factor = 10000. Then, using the "vst" method to find 2000 high variable genes (HVGs) by *FindVariableFeatures* function, these HVGs would be the features in the principal component analysis (PCA). Before PCA analysis (*RunPCA* function), the data was scaled using the *ScaleData* function. Clustering was done by using the *FindNeighbors* function, using the top 15 principal components (PCs) and the *FindClusters* function.

Cell type annotation was performed with *SingleR* (version 1.6.1)[71]. If a cluster co-expresses markers of two cell types, then the cluster likely contains double-droplets. Based on this, we excluded total 965 cells using the default parameters of the R package *DoubletFinder* (version 2.0) from three groups (P57, PL, and control)[72]. For visualization and analysis and to eliminate batch effects, the three groups were integrated as a Seurat object using Harmony (version 0.1.0)[73]. Then, the integrated object was clustered using FindClusters function with the *resolution* = 0.4. In addition, Glutamatergic neurons and GABAergic neurons were extracted and clustered to 39 subpopulations by setting the *resolution = 1.5*. We performed *FindAllMarkers* function to identify the marker gene of each cluster by setting cut-off: *min.pct = 0.1, logfc.threshold = 0.25, test.use = "wilcox"*. The DEGs between different groups was calculated by using FindMarker function, such as P57_vs_Control: *FindMarkers (object, ident.1 = "P57", ident.2 = "control")*.

To explore the potential function of marker genes in different cell populations or DEGs in different groups, we used R package *clusterProfiler (version 4.0.5)* to perform KEGG Function enrichment analysis. The *org.Mm.eg.db* (3.14.0) package was used to set the background gene list as all genes detected in sequencing. The Benjamin-Hochberg FDR method was used to adjust the right tail P-value (two-sided binomial statistical test).

## Statistics and reproducibility

The number of animals or samples used in each experiment are stated in the manuscript or in the figures. The numbers of experimental repetitions were as follows: Figs.1b, 3 times; Fig. 1c–f, once; Fig. 1g, twice; Fig. 1h-j, once; Fig. 2b, c, twice; Fig. 2d, e, twice; Figs. 2f, 3 times; Fig. 3a, twice; Fig. 3c, twice; Fig. 3e, f, twice; Fig. 3g, once; Fig. 3h–j, twice; Fig. S1a, once; Fig. S1b-e, twice; Fig. S2a–c, once; Fig. S3a–c, once; Fig. S4a and b, twice; Fig. S4c–e, once; Figs.S5a, 3 times; Figs. S5b, 4 times; Figs. S5c, 3 times; Fig. S5d and e, twice; Figs. S5f, 3 times; Fig. S5g, twice; Fig. S6a, twice; Fig. S6c, once; Fig. S6e–g, twice; Fig.S9a–h, twice; Fig.S9i and j, once; Fig. S10a–e, once; Fig. S10f–n, twice; Table S1, twice; Table S2, twice; Table S3, twice.

## Reporting summary

Further information on research design is available in the Nature Portfolio Reporting Summary linked to this article.

## Data availability

The mass spectrometry data generated in this study have been deposited in the iProX database consortium with the dataset ID IPX0005823000 under accession code: CdOj (URL: https://www.iprox. cn/page/PSV023.html;?url=1675073267636rwMF.). The snRNA-seq data generated in this study have been deposited in the Genome Sequence Archive (Genomics, Proteomics & Bioinformatics 2021) in National Genomics Data Center (Nucleic Acids Res 2022), China National Center for Bioinformation / Beijing Institute of Genomics, Chinese Academy of Sciences (GSA: CRA012207) (https://ngdc.cncb. ac.cn/gsa/browse/CRA012207). Source data are provided in this paper and can be found in the Source Data File. Source data are provided in this paper.

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

## Acknowledgements

We thank the instructive advice and support from Prof. Lan Ma and Feifei Wang from Fudan University, Prof. Jun O. Liu from Johns Hopkins University, and Prof. Wei Shen from the University of Shanghai for Science and Technology. This work was supported by the National Natural Science Foundation of China [22137002, 21877016 to Y.D.; 22031011, 21621002 to B.Y.; 91853205, 81625022 to C.L.; 81972621, 92253305 to W.J. and 81970727, 31900738 to L.X.], Science and Technology Commission of Shanghai Municipality (Grant 20JC1410900 to Y.D.), University Innovation Research Group in Chongqing (No. CXQT21016 to Y.D.), Chongqing Talent Program Project (No. CQYC20200302119 to Y.D.), High-Level Innovation Platform Cultivation Plan of Chongqing (to Y.D.), Joint Fund of the Natural Science Innovation and Development Foundation of Chongqing (to Y.D.), Program for Professor of Special Appointment (Eastern Scholar) at Shanghai Institutions of Higher Learning (to W.J.), the Fund of the State Key Laboratory of Bioorganic and Natural Products Chemistry [SKLBNPC16233] (to B.Y.), the Key Research Program of Frontier Sciences of the Chinese Academy of Sciences [ZDBS-LY-SLH030] (to B.Y.), Shanghai Municipal Science and Technology Major Project (No.2018SHZDZX01) (to L.X.), Shanghai Pujiang Program (19PJ1401800) (to L.X.), ZJLab, and Shanghai Center for Brain Science and Brain-Inspired Technology (to L.X.).

## Author contributions

Y.D. initiated the project, Y.D., B.Y., C.L., and W.J. designed and supervised the project; R.W., L.X., and G.B. did experiments and analyzed and prepared data; J.P., Q.M., and Z.W. analyzed the cell experiments and analyzed the data; D.Z., R.W., and C.X. contributed to molecular biological and animal experiments; C.P. and W.L. contributed to chemical synthesis; J.W. and J.H. contributed to enzyme activity assay and computational calculation; M.Z. and M.T. contributed to identify the target of protein by M.S.; G.B., G.W., L.G., Q.T., M.J., M.H., Y.W., Z.L. C.X., D.M., and C.L. contributed to neurobiological experiments; T.L. and Y. Z. contributed to MACO model; X.F. and J.L. contributed to HPLC experiments; the manuscript was written by R.W., G.B., L.X., W.J., and Y.D.

## Competing interests

The authors declare no competing interests.

## Additional information

[1]Key Laboratory of Metabolism and Molecular Medicine, Ministry of Education, Department of Biochemistry and Molecular Biology, School of Basic Medical Sciences, Shanghai Medical College, Fudan University, Shanghai, China. [2]State Key Laboratory of Medical Neurobiology and MOE Frontiers Center for Brain Science, Institutes of Brain Science, Shanghai Medical College, Fudan University, Shanghai, China. [3]Basic Medicine Research and Innovation Center for Novel Target and Therapeutic Intervention, Ministry of Education, Institute of Life Sciences, the Second Affiliated Hospital of Chongqing Medical University, Chongqing Medical University, Chongqing, China. [4]State Key Laboratory of Drug Research, Shanghai Institute of Materia Medica, Chinese Academy of Sciences, Shanghai, China. [5]State Key Laboratory of Bio-organic and Natural Products Chemistry, Shanghai Institute of Organic Chemistry, Chinese Academy of Sciences, Shanghai, China. [6]Department of Pharmacology, College of Pharmacy, Naval Medical University, Shanghai, China. [7]Lab of Tumor Immunology, Department of Human Anatomy, Histology and Embryology, Basic Medical School of Fudan University, Shanghai, China. [8]Department of Medicinal Chemistry, China Pharmaceutical University, Nanjing, China. [9]Department of Physiology and Pathophysiology, School of Basic Medical Sciences, Shanghai Medical College, Fudan University, Shanghai, China. [10]Department of Physiology, Cardiovascular Research Institute, University of California San Francisco, San Francisco, CA, USA. [11]These authors contributed equally: Ruina Wang, Lei Xiao, Jianbo Pan, Guangsen Bao. ✉e-mail: jiangw@fudan.edu.cn; cluo@simm.ac.cn; byu@mail.sioc.ac.cn; yjdang@cqmu.edu.cn

