## [Peer review file · Nature Communications]

REVIEWER COMMENTS

Reviewer #1 (Remarks to the Author):

Wang et al have investigated the effect of P57 (an oxypregnane steroidal glycoside isolated from *H. gordonii*) on body temperature. They show that P57 induces hypothermia and a torpor-like state. At molecular level P57 inhibits pyridoxal kinase (PDXK), leading to the accumulation of pyridoxal (PL) in the hypothalamus to cause hypothermia.

1. This manuscript shows very interesting and novel evidence about the role of P57 on thermoregulation. The study is of interest and the data mainly support the conclusions. However, it shows a major weakness in the investigation of the mechanism leading to P57-induced hypothermia: which are the reasons why P57-treated animals decrease body temperature: impaired BAT thermogenesis? decreased basal metabolic rate? vasodilation in the tail? is the sympathetic nervous system involved? etc. These questions need to be addressed, otherwise the presented evidence, although interesting is mainly descriptive.
2. In keeping with the previous points, the central mechanisms are also unclear: how does PL accumulation interfere with the hypothalamic mechanism/networks modulating energy balance? In this sense, there is too much focus on the MPA, while the c-fos responses in the BSTLD (Figure S8a-b) and the PVH (Figure S8a-c) seems much higher. This needs to be also addressed.
3. The effect of P57 on peripheral organs/tissues is morphologically evaluated in heart, kidney, liver, and muscle (Figure S1f). This kind of approach does not allow concluding lack of side-effects. Functional analysis (heart rate, blood pressure, kidney function and liver and muscle metabolism etc.) would require an evaluation.
4. PL values are given in % in Figure 3a. A proper quantification (as concentration in the appropriate units) should be provided.
5. MCAO model data are of extreme significance, and they should be better highlighted. In fact, they should be shown as regular Figure (rather than Supplemental) and they would provide an excellent manner to close the paper (last Figure), given the importance of the findings and their possible translational relevance.

6. In terms of data presentation, I personally find difficult to follow some of the graphs, because of the similar color tones chosen. For example, in Figure 1 the blue scale seems to close. The same in Figure 3e, where I would recommend splitting WT and KO data in two different graphs. Also, the green tones in Figure 3g are very similar.

Reviewer #2 (Remarks to the Author):

Wang et al have a comprehensive study of the impact of P57 on the mouse. P57 falls in a long line of other compounds that cause hypothermia in the mouse – those include H₂S, AMP, CHA, NPY just to name a few. The mouse is particularly tolerant to bouts of hypothermia, so the discovery of a new compound that can inhibit metabolism is not particularly exciting. However, the authors' work goes much further than the studies of these other compounds, and propose a central mechanism within the POA of the hypothalamus. The problem here is that P57 could be acting directly at the level of the hypothalamus, or it could be acting as a non-specific inhibitor of Vitamin B dependent pathways, or, as I suspect, in both ways. The dream experiment would be to somehow block entry of P57 into the cerebral spinal fluid. That way, the peripheral effects (likely non-specific on metabolic rate) can be dissociated from the specific effects centrally. What is missing in this manuscript is any sort of direct comparison to natural torpor. Natural torpor is easy to induce – an overnight fast or a few days of caloric restriction. Without that comparison, we can't be sure that the effect of P57 is not just one of simple metabolic inhibition that forces the metabolic rate to slow, with the ensuing hypothermia. One very simple measure is to calculate the rate of temperature loss during entrance or gain during recovery. The rate of temperature loss in natural torpor is much slower than with exogenous compounds (like CHA, for example). And the rate of temperature gain is much faster in recovery from natural torpor than after receiving metabolic inhibitors. Given that P57 impacts the active form of Vit B6, and this vitamin is a cofactor for 160 enzymes, one could easily assume that P57 is working simply to inhibit enzymatic activity, and that slows down overall metabolic rate in this hypothermia tolerant species. Indeed, maybe half of the hypothermia is independent of PDXK, if I am reading Figure 3e correctly. If P57 is truly acting through the hypothalamus to induce hypothermia like torpor, and not working peripherally, then these rates of temp gain/loss should be similar between natural torpor and P57 administration. There are plenty of examples in the literature where these rates were measured in mice. I should say that the authors should be commended for using qualifying terms, such as "hibernation-like" or on Line 800 "caused a hypothermic and hypometabolic state in a reversible manner". However, it is impossible to make any important conclusions on specific metabolic inhibition or thermoregulation without the experiment of peripheral delivery with no central delivery. The authors show that P57 centrally CAN cause hypothermia (with whopping doses of 1 mg/kg range, I might add) ... but this doesn't mean that this is how P57 ACTUALLY works. In the absence of such an experiment, the authors must bring up this important distinction between specific/central vs. non-specific peripheral as a real possibility. Any comparisons to natural torpor also are important to their conclusions.

Specific comments

Line 41 – “recuperated” should be “blunted” or some other word

Line 71 – it should be stated here that the P57 was given intraperitoneally (as it is stated in Figure 1 legend)

Line 91 – remove the word “therapeutic” as it does not belong here.

Figure S1F – what was the timing of when the tissue was taken relative to the injections; and/or what was the temperature of the mouse?

Figure 5F,G. I think these measurements were recorded with an IR camera, in which case the authors should not report core temperature (for example, on the Y-axis) but rather surface temperature.

Line 103 – Bio-P57 is introduced here, but without its definition, which I assume means biotinylated.

Line 219-221. I don't agree that the consecutive treatment with P57 leaves the thermoregulatory system functional. It very well could be, and should be shown otherwise, that P57's effects are at least partially mediated as a non-specific inhibition of enzymatic activity where the thermoregulatory system is abandoned. To determine if the thermoregulatory system is functional, one can lower the ambient temperature. If body temperature stays constant, then I would believe the thermoregulatory system is functional. If body temp drops, then this is evidence that either the defended minimum T_b is not reached, or the mouse is not responding to the drop in body temp.

Reviewer #3 (Remarks to the Author):

The authors describe the significant discovery that systemic or brain administration P57, an oxypregnane steroidal glycoside with the prominent activity of appetite suppression, can induce a significant hypothermia and hypometabolism in mice and rats. Further, the authors' experiments provide strong evidence that these thermoregulatory effects of P57 are mediated via inhibition of PDXK, resulting in an increase in brain PL, which, when injected into the POA, can also induce a hypothermia. The authors also show that treatment with P57 can reduce the infarct volume in a MCAO model of stroke. Together, these are exciting and potentially translational data supporting the application of P57 for the induction of a therapeutic hypothermia to reduce cellular damage from ischemic insult. The importance of these results is overshadowed by the second part of the manuscript in which the authors attempt to identify the neuronal circuit mechanism in the POA through which P57, reduced PDXK or increased PL can produce hypothermia. Although c-Fos responses during the induced hypothermia are a useful beginning to such an investigation, the remaining series of experiments, particularly those involving detailed classifications of neurons with GABAergic and glutamatergic markers, are poorly designed and the results are uninterpretable. Thus, we are left with a manuscript half of which reveal

novel and highly relevant discoveries of potential clinical interest, and half of which is minimally useful information that sheds little light on the neuronal mechanisms through which this effect is mediated.

Despite the use of several techniques, the study lacks several important control experiments to sustain some of the proposed conclusions. First, the authors have provided no definition of 'torpor' or a 'torpor-like state', thus, their conclusions regarding the parallels between the P57-induced state and natural torpor (or hibernation) remain subjective. Second, there is no clear demonstration that POA neurons are uniquely responsible for the P57-mediated hypothermia. Third, it is not clear whether the reduction in stroke volume is mediated by the hypothermia or by direct action P57 on brain cells.

Major

1. The authors should eliminate their repeated claims that P57 induces a hibernation-like/torpor-like state and describe the effects of P57 administration simply as inducing a hypothermia and hypometabolism. These are the principal effects that are relevant to their intention to develop an approach to inducing therapeutic hypothermia, and they have not demonstrated that the P57-induced state truly mimics natural torpor. In particular, they have not addressed the question of whether blocking or altering the P57 mechanism of action prevents entrance into natural torpor, or whether the P57-induced state mimics the alterations in EEG pattern and in cardiovascular function that accompany natural torpor? Furthermore, the authors use the terms, hibernation, and torpor, synonymously, suggesting some confusion about the definition of these terms. Line 82 A long-lasting hypothermic and hypometabolic state is similar to hibernation or daily torpor 32. This is absolutely not true. Anesthesia, for instance, causes hypothermia and a hypometabolic state, but no one considers this to be a torpor or a hibernating state. Several additional physiological features define a state of torpor or a state of hibernation and these were not considered by the authors. The states of hibernation and torpor exhibit several different characteristics. As an example, mice do not hibernate but are able to enter torpor.

2. The conclusions regarding neuronal mechanisms underlying the P57 hypothermia that was derived from the experiments regarding POA neurons are simply speculation and, most importantly, the authors do not propose any precise circuit mechanism for the role of POA in the P57 induced hypothermia. Despite a nearly global c-Fos activation of the POA, the authors focus their attention on the MPA nucleus on the only basis that previous work has shown a thermoregulatory role in these areas. None of the experiments in this manuscript were designed to clearly demonstrate that "MPA is a core target of P57 for inducing hypothermia". The experiment in POA PDXK knockout mice should have demonstrated the role of POA in hypothermia induction, however, it is unclear why the knockout of PDXK did not induce a hypothermia comparable to that induced by P57.

3. The authors seem unaware of the complexity of the POA circuits controlling thermogenesis as is clear from the lack of citations to important literature in this field. It seems that P57 is acting on both glutamatergic and GABAergic neurons in POA. Considering that, POA controls many homeostatic functions it is unclear from the experiments proposed whether the activation of these neurons are directly responsible for the hypothermia or they mainly control other functions (hypophagia, sleep...).

4. Line 76-79 Authors claim that motor behavior of P57-treated mice returns to normal 200 min following administration. This seems a misinterpretation of the data. First, important data about

baseline motor activity are missing. Do these animals show normal motor activity before treatment? Was the experiment run at the same time of the day for each animal? light or dark phase? Second, it seems the vehicle is causing an increase in motor activity that normalizes over time, returning to the same level observed in P57-treated animals. Without baseline motor activity data, the authors cannot rule out any conclusion.

5. Line 89- 95 Author suggest here, and in the discussion section (223- 225) that P57 has neuroprotective effects in stroke. However, a key control experiment to determine whether the neuroprotective effect is due to an effect of P57 on ischemic tissue or simply to the P57-induced hypothermia is missing. What is the effect on infarct volume if the MCAO animals were given P57, but maintained at normothermia? The discussion should reflect the lack of this important control. It is certainly interesting that P57 works in rats, a species, like humans, that is unable to naturally express torpor. However, the mechanism of action for the induction of hypothermia could be very different between rats and mice. In this regard, many recent studies on thermoregulation highlighted the thermoregulatory mechanism differences in POA between these two species. Authors should add this caveat in the discussion and clarify why these data were obtained in a different species.

6. Line 150. Authors stated that PL level increased in the hypothalamus but no obvious change was observed in whole brain tissues. The lack of effect could be the result of a washout effect due to a higher sampling volume. The same sample volume of other brain areas would have been a better control. In addition, why were the c-Fos data not reported for other brain areas besides the POA – did neurons in no other brain areas exhibit increased c-Fos? This would have also offered an indication of the site of action of PL or P57. The study focuses mostly on POA on the basis that other studies have found these areas to be important for induction of hypothermia. However, other thermoregulatory areas (e.g., medullary and pontine structures) should have been assessed to determine if the POA is the only site of action of P57. The experiment conducted with AAV suggests otherwise that POA is probably not the single site of action, as IP injection in POA PDXK knockout mice only reduces, instead of completely preventing, P57 induced hypothermia. The possibility of the presence of a different site of action is also supported by the high c-Fos expression in PVH, a brain area also involved in the modulation of core body temperature. Why did the authors not directly inject P57 in PVH?

7. Line 163 injection of P57 in POA. The volume of the injection and a clear description of the method used for injection into POA must be reported. Additionally, the size of the injection sites must be assessed from postmortem histology and plotted on atlas drawings of the mouse hypothalamus. A figure showing a histological example of the injection site and its extension into POA, for both P57 and AAV, should be added to Fig 3.

8. Line 198-212 None of the data obtained here are supporting the idea that “MPA is a core target of P57 for inducing hypothermia”. Keys control experiments are missing to claim this outcome.

- Fiber photometry data: given the large injection in POA it is unclear how the authors can claim that the recorded fluorescence is coming solely from MPA neurons. Also, the placement of the recording probe is determined by the cessation of further increases in fluorescence intensity which means that if the injection x-ires-Cre was misplaced, then the recording probe was also misplaced. Fig 3e clearly shows

the variability in the positions of the injection sites. Furthermore, no group data and statistical comparison are provided for these results (Fig 5f, g).

- Fiber optometry could produce a local temperature increase. MPA contains warm sensitive neurons that in response to warm stimulus could induce inhibition of thermogenesis and reduction in body temperature. No controls are provided for this very real possibility to explain the results.
- Control data of the firing rate of MPA neurons in response to the vehicle are missing. This must be provided. Also what vehicle was used?
- PVH shows a large C-fos activation (larger than MPA) in response to P57 administration, however, no experiments were proposed to assess the role of this nucleus which is also involved in thermoregulation (PMID: 19129373). A higher magnification photomicrograph of the PVH c-Fos results must be provided – from the low power photo, it appears that the entire PVH is c-Fos positive.
- “In addition, the fluctuation in the degree of GCaMP also increased with the reduction of core body temperature (Fig.5f, g and Fig.S9d, e).” how this was assessed? Group data and statistical comparison must be provided.

9. Line 219-225 “consecutive treatment of P57 maintains the lengthy hypothermic state while the thermoregulatory system remains functional when the theoretical core temperature is lowered.” It is not clear on what basis the authors can state that the thermoregulatory system remains functional. None of the proposed experiments have assessed this. “there is no tissue damage” not clear how the tissue damage was assessed. Histological sections showing ‘damage’ should be provided and a clear description should be added in the methods section.

10. Line 219-225 Based on the following statement:

- First, consecutive treatment of P57 maintains the lengthy hypothermic
- Second, there is no tissue damage despite
- Third, animal is able to recover from P57-induced hypothermia

Authors state that “Hypothermic state induced by P57 shares several features with hibernation-like state and therapeutically induced hypothermia”. I am not sure this is a good way to define a hibernation-like state, in fact, anesthetic does just the same but is not considered to induce a state of either hibernation or torpor.

Fig. 1 the title of the figure should reflect what is represented in the figures and on which species, in this case just the hypothermia in mice. “P57 induces a reversible hypothermia in mice”

Fig. 3 Add a panel showing an example of injection sites for AAV and P57 in POA similar Fig. 5e

Fig. 5 a panel comparing the GABA non-GABA neuronal activity in response to the vehicle should be reported. Group data for panels f & g are missing.

Fig. S1 “Administration of P57 induces hypothermia...” however the figure does not show hypothermia but hypometabolism. The title should reflect what the figure shows.

Latency to fall, at recovery, following 200 min after P57 administration, seems extraordinarily perfect. However, from panel a it seems that, following 200 min after P57 administration, the animal has not yet recovered from the hypometabolic effect of the drugs. This seems to be true also for the locomotor activity in panel d.

Baseline locomotor activity is missing? This should be reported. Panel d seems to suggest an effect of the vehicle (maybe stress due to injection) more than a reduced motor activity induced by P57. The baseline motor activity is fundamental to show the reduction in motor activity that is claimed by the authors. Why latency to fall for the vehicle is not reported? This must be included.

Panel f. there are several informations missing that must be reported in the methods section. Time of euthanasia following injections of P57, description of the staining, and the method used to determine that there are no pathological changes (cell dimensions? Lack of presence of particular structures?).

Fig. S5 baseline motor activity should be reported in panel e. Compare the latency to fall variability between fig.S5 and S1. Why in S1 there is no variability at time 0 ad time 200? Why latency to fall for the vehicle is not reported? this must be included.

Minor

In the methods section and in the relevant place in the result section, add more details on intraparenchymal injection procedures, including volume of injection, concentration, and type of vehicle used for each drug.

Line 80 should add hypophagic!! as the effects could be due to something like a fall in feeding, leading to fall in blood glucose, and maybe torpor induction

Line 209 Check reference to figures. (Fig.5e-g.....) should probably be (Fig.5f-g.....).

Fig. S10 A schematic of the P57 mechanism of action should go in the main manuscript.

Reviewer #4 (Remarks to the Author):

This is potentially interesting study, where authors identified targeted pharmacological compound P57 with ability to induce torpor or hibernation-like states in rodents. Despite important clinical significance of this topic, authors do not have enough experimental support to make this statement.

Major points:

1. Torpor is characterized by periods of complete inactivity, reduction in metabolic rate that precedes reduction in body temperature (below 31C). Behavioral experiments (rotarod and open-field) (Fig.S1) strongly indicates that authors are talking about pharmacological induction of hypothermia and hypometabolism, but NOT torpor or hibernation-like state.
2. It would be useful to perform blood comprehensive metabolic panel to measure: glucose (during hibernation glucose is relatively stable); beta-hydroxybutyrate (significantly 2-3 times increased during hibernation), insulin, etc...
3. It is not clear what histological analysis (Fig. S1f) is telling us and what conclusions authors can make from it. Perhaps, it would be useful to perform staining for apoptotic markers, etc
4. It is not clear why hypothermic phenotype and mice and rats are so different upon P57 administration. In mice CBT drops to 26C; in rats only to 34C. One explanation is that mice have an ability to undergo natural torpor during fasting, but rats do not. Authors should discuss this species-specific phenotype. Moreover, authors conclude that "P57 can induce hibernation-like state in rodents". I do not think that authors provide strong experimental support for this statement for mice and even less so for rats.
5. Fig. 3 Why CBT phenotype is so different from Figure 1. Max drop only to 30C. Do authors observed changes in metabolic rate similar to Fig. 1
6. Fig. S5 Behavioral data does not support the statement that animals are going into torpor or hibernation-like state. In case of mouse torpor activity is reduced almost to zero like sleep state. It is hard for me to imagine rotarod experiments during sleep.

Reviewer #5 (Remarks to the Author):

In the manuscript “Natural Product P57 Induces Hibernation-like State through Targeting Pyridoxal Kinase” by Wang et al. the effect and function of the natural product P57 on mice and rats is monitored. The authors observe by chance that P57 administration lowers body temperature and metabolic rates. To address a potential mechanism-of-action the authors perform affinity purifications using biotinylated P57 as bait and identify PDXK as potential interactor. In in vitro analyses and structural studies, the mode of binding of P57 to PDXK is determined. P57 appears to be an inhibitor of PDXK competing with PL. Continuing with in vivo analyses the brain area in which P57 functions is determined. Using single-cell RNAseq potential target cells are identified. Finally, responses of neurons to P57 and PL in the MPA are monitored.

This is an interesting paper which spans from vitro analyses determining drug-protein interactions to in vivo studies analysing phenotypic responses of rodents to drug administration. The findings are of medical/clinical interest as induction of hibernation-like states represents an interesting treatment strategy to acute insults affecting e.g. brain or heart metabolism. However, several points should be addressed prior publication to ensure the soundness of presented data. Major points are:

- (1) How were potential binding partners shortlisted in the MS-based proteomics studies? This section is kept very short and the reader cannot follow the selection criteria. It is not clear how many proteins were identified and why GSTP1 and PDXK were of specific interest.
- (2) Along this line, the proteomic data should be presented “as a whole” including the shortlisted proteins, e.g. using a volcano plot, before validation experiments coupling IPs to western blots are performed.
- (3) I did not find Table S1 accompanying the manuscript. This table is crucial to be able to evaluate proteomic data.
- (4) In the methods section it states “Affinity chromatography experiment and iTRAQ analysis”; however, no experimental details and methods are listed that refer to the iTRAQ labeling experiments. Thus, it is not clear if iTRAQ was used and when it was used how the ratios were treated and binders were determined.
- (5) MS raw data should be deposited on a public server such as PRIDE DB.

Minor points:

- (1) The manuscript should be spell- and grammar-checked by a native speaker.
- (2) The introduction is very short. For non-experts it would be helpful to know why the hypothalamus is critical for body temperature control.

Reply to Reviewer #1:

*Wang et al have investigated the effect of P57 (an oxypregnane steroidal glycoside isolated from *H. gordonii*) on body temperature. They show that P57 induces hypothermia and a torpor-like state. At molecular level P57 inhibits pyridoxal kinase (PDXK), leading to the accumulation of pyridoxal (PL) in the hypothalamus to cause hypothermia.*

1. This manuscript shows very interesting and novel evidence about the role of P57 on thermoregulation. The study is of interest and the data mainly support the conclusions. However, it shows a major weakness in the investigation of the mechanism leading to P57-induced hypothermia: which are the reasons why P57-treated animals decrease body temperature: impaired BAT thermogenesis? decreased basal metabolic rate? vasodilation in the tail? is the sympathetic nervous system involved? etc. These questions need to be addressed, otherwise the presented evidence, although interesting is mainly descriptive.

Response: Thank you for the constructive comments. To explain the mechanism of P57-induced hypothermia, we monitored the temperature of BAT and tail skin with IR digital thermographic camera. Infrared thermal images indicated that the temperature of BAT and tail skin decreased significantly after P57 treatment, and the temperature of BAT decreases earlier than that of tail skin (Review Figure 1-1). Thus, these results indicate that P57 mainly suppresses heat production to induce hypothermia. In manuscript, we have demonstrated that P57-treated animals significantly decreased basal metabolic rate.

Using biotin labeled P57, we found PDXK was the target of P57 and validated it *in vivo*. Mice with PDXK knockout in MPA demonstrated lower core temperature compared with the control animals in a reduced ambient temperature. Moreover, locally administered P57 into MPA, not BSTLD or PVH, rapidly induced hypothermia. Using snRNA-seq, patch-clamp technique and fiber photometry calcium signal recording, we found that P57 and PL increase the activity of MPA GABA and glutamate neurons to reduce core body temperature. Taken together, our findings demonstrate that P57 targets PDXK to activate MPA neurons to induce hyperthermia and hypometabolism.

Review Figure 1-1 P57 suppresses heat production. Surface temperature of BAT (a) and tail (b) of P57-treated mice.

2. In keeping with the previous points, the central mechanisms are also unclear: how does PL accumulation interfere with the hypothalamic mechanism/networks modulating energy balance? In this sense, there is too much focus on the MPA, while the c-fos responses in the BSTLD (Figure S8a-b) and the PVH (Figure S8a-c) seems much higher. This needs to be also addressed.

Response: We appreciate for your suggestion. The analysis of snRNA-seq data suggests that the neurons in MPA was significantly affected by PL and P57 treatment. To evaluate which area has a major role in inducing hypothermia, we directly injected 150µg/kg of P57 into MPA, BSTLD and PVH respectively. The core temperature of mice decreased significantly after injection of P57 directly into MPA, but didn't change much directly into BSTLD and PVH (Review Figure 1-2). Thus, it is strongly suggested that MPA plays a major role for P57-induced hypothermia. We have added these results into Fig.5c and Fig. S9d, e of the revised manuscript.

Review Figure 1-2 P57 induces hypothermia mainly by working on MPA. Core temperature of mice treated with P57 in the MPA (a), PVH (b) or BSTLD (c). P57 (150 µg/kg) or the vehicle control was directly injected into the subregion of hypothalamus.

3. The effect of P57 on peripheral organs/tissues is morphologically evaluated in heart, kidney, liver, and muscle (Figure S1f). This kind of approach does not allow concluding lack of side-effects. Functional analysis (heart rate, blood pressure, kidney function and liver and muscle metabolism etc.) would require an evaluation.

Response: Thank you for your valuable suggestion. We have added the function analysis, including heart rate and blood pressure according to the reviewer's suggestion. As shown in Review Figure 1-3, after administrated 4 consecutives i.p. of P57 to mice, heart rate firstly slowed down, then recovered. However, the blood pressure was hardly changed. We have added these results into Fig.S2a, b of the revised manuscript.

Review Figure 1-3 Blood pressure and heart rate measurements of consecutive administration of P57. 25.0 mg/kg of P57 or the vehicle control was injected intraperitoneally into mice every 3 h for 4 times. (a) Blood pressure of P57-treated mice. (b) Heart rates of P57-treated mice.

4. PL values are given in % in Figure 3a. A proper quantification (as concentration in the appropriate units) should be provided.

Response: Thank you for your suggestion. I have changed title of Y-axis with concentration of PL content (Review Figure 1-4). We have updated the figures in the revised manuscript.

Review Figure 1-4 LC-MS detection of PL in hypothalamus (a) and brain tissues (b).

5. MCAO model data are of extreme significance, and they should be better highlighted. In fact, they should be shown as regular Figure (rather than Supplemental) and they would provide an excellent manner to close the paper (last Figure), given the importance of the findings and their possible translational relevance.

Response: Thank you for your valuable suggestion. We have evaluated P57 effect on mice and found that P57 still played the protective role in mice MCAO model (Review Figure 1-5). Since the manuscript was focused on mice, we have added mice MCAO result in main Figure (Fig.1f and Fig. S3a, b of the revised manuscript) according to the reviewer's suggestion.

Review Figure 1-5 P57 has a neuroprotective effect in MCAO model. P57 (25.0 mg/kg) or the vehicle was injected intraperitoneally into mice 2 h after middle cerebral artery occlusion. Core temperature of mice in MCAO + P57+ Warming group maintained by the heating pad. (a) Core temperature of mice in MCAO model. (b) Representative images of TTC staining of brain slices in MCAO model treated with P57. (c) Statistic infarct volumes of brain slices from mouse model of MCAO.

6. In terms of data presentation, I personally find difficult to follow some of the graphs, because of the similar color tones chosen. For example, in Figure 1 the blue scale seems to close. The same in Figure 3e, where I would recommend splitting WT and KO data in two different graphs. Also, the green tones in Figure 3g are very similar.

Response: Thank you for your suggestion. I have changed the similar color tones in terms of data presentation in revised manuscript. In Figure 3e of the revised manuscript, I have split WT an KO data into two different panels.

Reply to Reviewer #2

Wang et al have a comprehensive study of the impact of P57 on the mouse. P57 falls in a long line of other compounds that cause hypothermia in the mouse – those include H₂S, AMP, CHA, NPY just to name a few. The mouse is particularly tolerant to bouts of hypothermia, so the discovery of a new compound that can inhibit metabolism is not particularly exciting. However, the authors' work goes much further than the studies of these other compounds, and propose a central mechanism within the POA of the hypothalamus. The problem here is that P57 could be acting directly at the level of the hypothalamus, or it could be acting as a non-specific inhibitor of Vitamin B dependent pathways, or, as I suspect, in both ways. The dream experiment would be to somehow block entry of P57 into the cerebral spinal fluid. That way, the peripheral effects (likely non-specific on metabolic rate) can be dissociated from the specific effects centrally. What is missing in this manuscript is any sort of direct comparison to natural torpor. Natural torpor is easy to induce – an overnight fast or a few days of caloric restriction. Without that comparison, we can't be sure that the effect of P57 is not just one of simple metabolic inhibition that forces the metabolic rate to slow, with the ensuing hypothermia. One very simple measure is to calculate the rate of temperature loss during entrance or gain during recovery. The rate of temperature loss in natural torpor is much slower than with exogenous compounds (like CHA, for example). And the rate of temperature gain is much faster in recovery from natural torpor than after receiving metabolic inhibitors. Given that P57 impacts the active form of Vit B6, and this vitamin is a cofactor for 160 enzymes, one could easily assume that P57 is working simply to inhibit enzymatic activity, and that slows down overall metabolic rate in this hypothermia tolerant species. Indeed, maybe half of the hypothermia is independent of PDXK, if I am reading Figure 3e correctly. If P57 is truly acting through the hypothalamus to induce hypothermia like torpor, and not working peripherally, then these rates of temp gain/loss should be similar between natural torpor and P57 administration. There are plenty of examples in the literature where these rates were measured in mice. I should say that the authors should be commended for using qualifying terms, such as "hibernation-like" or on Line 800 "caused a hypothermic and hypometabolic state in a reversible manner". However, it is impossible to make any important conclusions on specific metabolic inhibition or thermoregulation without the

experiment of peripheral delivery with no central delivery. The authors show that P57 centrally CAN cause hypothermia (with whopping doses of 1 mg/kg range, I might add) ... but this doesn't mean that this is how P57 ACTUALLY works. In the absence of such an experiment, the authors must bring up this important distinction between specific/central vs. non-specific peripheral as a real possibility. Any comparisons to natural torpor also are important to their conclusions.

Response: We appreciate your constructive comments. The Reviewer mainly concerned about two questions: 1) Whether P57 caused hypothermia by acting at hypothalamus, periphery, or both; 2) Is it reasonable to define the P57-induced hypothermia as torpor or hibernation-like.

Firstly, we agree with the Reviewer that P57 may induce hypothermia by both peripheral and central effects. As the Reviewer mentioned, blocking the entry of P57 into the cerebral spinal fluid is the dream experiment to discriminate the central and peripheral effects. However, this experiment seems to be impossible to be conducted. In this study, we focused on the possible target of P57 in central nervous system, and proved that MPA is a candidate region by several experiments: 1) all the results from c-Fos immunostaining, electrophysiological recording, and fiber photometry recording showed that P57 increased the activity of MPA neurons; 2) local microinjection of P57 into MPA, but not into PVH or BSTLD, induced the hypothermia, which is similar as the effect of intraperitoneal injection of P57; 3) knocking down PDXK in MPA induced more core temperature drop when lowering ambient temperature and also significantly reduced the P57-induced hypothermia. Since knocking down PDXK in MPA with AAV viruses did not totally block the P57-induced hypothermia, we speculate that this may due to the efficiency of virus infection or the contribution of the peripheral effects. We have added these statements in the Discussion in the revised manuscript (Page 10, Line 230~239).

Secondly, comparing the P57-induced hypothermia with natural torpor, P57 can induce hypothermia in minutes, which is different to the torpor induced by fasting. To further characterize the hypothermia induced by P57, we conducted a new experiment to measure the change of temperature during the torpor induced by fasting as you

suggested (Review Figure 2-1). Though the rate of temperature loss during entrance was similar between P57-induced hypothermia and the fasting-induced torpor, the rate of temperature gain during recovery was slower for the P57-induced hypothermia. Based on several Reviewers' comment and our new experiment, we used hypothermia to instead of torpor or hibernation-like in the revised manuscript. We certainly believe that P57 will be served as a valuable tool for future to further discriminate the torpor, hibernation-like or hypothermia.

Review Figure 2-1 Comparison of mouse core temperature change during fasting-induced torpor (FIT) and P57-induced hypothermia.

Specific comments

Line 41 – “recuperated” should be “blunted” or some other word

Response: We have changed “recuperated” into “blunted” in the revised manuscript.

Line 71 – it should be stated here that the P57 was given intraperitoneally (as it is stated in Figure 1 legend)

Response: We have stated P57 was given intraperitoneally in the revised manuscript.

Line 91 – remove the word “therapeutic” as it does not belong here.

Response: The “therapeutic” has been removed.

Figure S1F – what was the timing of when the tissue was taken relative to the injections; and/or what was the temperature of the mouse?

Response: The tissues were taken after 72 hours relative to the first injection. The core temperature of mice has been normal after 60 hours relative to the first injection as shown in Review Figure 2-2. At 72 hours relative to the first injection, the core temperature of mice were 36.38 ± 0.27 °C. We have added these details in the figure legend.

Review Figure 2-2 Core temperature of mice treated with P57 for four times. P57 (25.0 mg/kg) or the vehicle control was injected intraperitoneally into mice at 0, 3, 6 and 9 hours (arrow). Core temperature (TC) was measured and recorded by Anilogger® core temperature monitoring system every 15 minutes, **** P < 0.0001.

Figure 5F,G. I think these measurements were recorded with an IR camera, in which case the authors should not report core temperature (for example, on the Y-axis) but rather surface temperature.

Response: Yes, these measurements were recorded with an IR camera. We have replaced title of Y-axis with T_s (T_s means surface temperature of mice).

Line 103 – Bio-P57 is introduced here, but without its definition, which I assume means biotinylated.

Response: Yes, Bio-P57 means biotinylated probe of P57. We have added the definition.

Line 219-221. I don't agree that the consecutive treatment with P57 leaves the thermoregulatory system functional. It very well could be, and should be shown otherwise, that P57's effects are at least partially mediated as a non-specific inhibition of enzymatic activity where the thermoregulatory system is abandoned. To determine if the

thermoregulatory system is functional, one can lower the ambient temperature. If body temperature stays constant, then I would believe the thermoregulatory system is functional. If body temp drops, then this is evidence that either the defended minimum T_b is not reached, or the mouse is not responding to the drop in body temp.

Response: Thanks for your comment. We intended to emphasize that mouse core temperature can still back to normal after consecutive treatment with P57. As shown in the Review Figure 2-3, we also conducted the new experiment as the Reviewer suggested, and found that P57 induced more mouse core body temperature drop at 10°C than that at room temperature (25°C). That's reasonable because P57 suppresses heat production with a lower metabolic rate to induce hypothermia, and the lower temperature (10°C) exacerbates the imbalance between heat production and heat dissipation. Therefore, “the consecutive treatment with P57 leaves the thermoregulatory system functional” is not correct, and we have deleted it in the revised manuscript.

Review Figure 2-3 P57-induced mouse core temperature change with different ambient temperatures. Mice were adapted for 2 hours at 10°C before given P57 intraperitoneally, and then maintained another 4 hours at 10°C.

Reply to Reviewer #3:

The authors describe the significant discovery that systemic or brain administration P57, an oxypregnane steroidal glycoside with the prominent activity of appetite suppression, can induce a significant hypothermia and hypometabolism in mice and rats. Further, the authors' experiments provide strong evidence that these thermoregulatory effects of P57 are mediated via inhibition of PDXK, resulting in an increase in brain PL, which, when injected into the POA, can also induce a hypothermia. The authors also show that treatment with P57 can reduce the infarct volume in a MCAO model of stroke. Together, these are exciting and potentially translational data supporting the application of P57 for the induction of a therapeutic hypothermia to reduce cellular damage from ischemic insult. The importance of these results is overshadowed by the second part of the manuscript in which the authors attempt to identify the neuronal circuit mechanism in the POA through which P57, reduced PDXK or increased PL can produce hypothermia. Although c-Fos responses during the induced hypothermia are a useful beginning to such an investigation, the remaining series of experiments, particularly those involving detailed classifications of neurons with GABAergic and glutamatergic markers, are poorly designed and the results are uninterpretable. Thus, we are left with a manuscript half of which reveal novel and highly relevant discoveries of potential clinical interest, and half of which is minimally useful information that sheds little light on the neuronal mechanisms through which this effect is mediated.

Despite the use of several techniques, the study lacks several important control experiments to sustain some of the proposed conclusions. First, the authors have provided no definition of 'torpor' or a 'torpor-like state', thus, their conclusions regarding the parallels between the P57-induced state and natural torpor (or hibernation) remain subjective. Second, there is no clear demonstration that POA neurons are uniquely responsible for the P57-mediated hypothermia. Third, it is not clear whether the reduction in stroke volume is mediated by the hypothermia or by direct action P57 on brain cells.

Response: Thanks for the Reviewer's positive comment and questions raised. We have conducted new experiments and data analyses based on your suggestion to improve the quality of our manuscript, as stated below.

Major

1. *The authors should eliminate their repeated claims that P57 induces a hibernation-like/torpor-like state and describe the effects of P57 administration simply as inducing a hypothermia and hypometabolism. These are the principal effects that are relevant to their intention to develop an approach to inducing therapeutic hypothermia, and they have not demonstrated that the P57-induced state truly mimics natural torpor. In particular, they have not addressed the question of whether blocking or altering the P57 mechanism of action prevents entrance into natural torpor, or whether the P57-induced state mimics the alterations in EEG pattern and in cardiovascular function that accompany natural torpor? Furthermore, the authors use the terms, hibernation, and torpor, synonymously, suggesting some confusion about the definition of these terms. Line 82 A long-lasting hypothermic and hypometabolic state is similar to hibernation or daily torpor 32. This is absolutely not true. Anesthesia, for instance, causes hypothermia and a hypometabolic state, but no one considers this to be a torpor or a hibernating state. Several additional physiological features define a state of torpor or a state of hibernation and these were not considered by the authors. The states of hibernation and torpor exhibit several different characteristics. As an example, mice do not hibernate but are able to enter torpor.*

Response: Thanks for your suggestion. We agree with the Reviewer that we should cautiously claim the hibernation-like/torpor-like state. We conducted a new experiment to induce natural torpor with fasting, and found that the rate of core temperature loss was similar between fasting-induced torpor and P57 administration, but the rate of temperature gain was different (Review Figure 3-1 a). Though the metabolic rate was reduced by around 30% (Review Figure 3-1 b) and the core temperature declined to be around 31°C after intraperitoneal injection of P57, which are consistent with torpor-like state as previous studies reported, we have changed our statement and described the effects of P57 administration as inducing a hypothermia and hypometabolism in the revised manuscript.

Review Figure 3-1 (a) comparison of mouse core temperature change during fasting-induced torpor (FIT) and P57-induced hypothermia. (b) Carbon dioxide output level of P57-treated mice.

2. The conclusions regarding neuronal mechanisms underlying the P57 hypothermia that was derived from the experiments regarding POA neurons are simply speculation and, most importantly, the authors do not propose any precise circuit mechanism for the role of POA in the P57 induced hypothermia. Despite a nearly global c-Fos activation of the POA, the authors focus their attention on the MPA nucleus on the only basis that previous work has shown a thermoregulatory role in these areas. None of the experiments in this manuscript were designed to clearly demonstrate that “MPA is a core target of P57 for inducing hypothermia”. The experiment in POA PDXK knockout mice should have demonstrated the role of POA in hypothermia induction, however, it is unclear why the knockout of PDXK did not induce a hypothermia comparable to that induced by P57.

Response: Thanks for the Reviewer’s comment. Based on our previous results and new conducted experiments, our study supports the MPA is the core target of P57 for inducing hypothermia: 1) intraperitoneal injection of P57 and direct application of P57 are able to activate MPA neurons, but P57 did not change the cFos expression in the vLPO (Review Figure 3-2 a); 2) microinjection of P57 into MPA, but not into PVH or BSTLD, is sufficient to reduce mouse core temperature, which is similar as the effect of intraperitoneal injection of P57 (Review Figure 3-2 b-d); 3) knockout PDXK in MPA, as checked with virus expression, can significantly reduce the P57-induced hypothermia. We also

conducted new experiment to monitor mouse core temperature change with ambient temperature after knocking out PDXK in MPA, and observed that reducing ambient temperature induced more temperature drop in the PDXK knockout mice (Review Figure 3-2 e). Therefore, our results support that MPA is the core target of P57 to induce hypothermia. We have added these comments in Discussion section of the revised manuscript (Page 10, Line 230-240).

Review Figure 3-2 P57 targets MPA, but not vLPO, PVH, or BSTLD to induce hypothermia. (a) Quantification of c-Fos-positive neurons in the vLPO (left) and brain sections containing vLPO immunostained with a neuronal activation marker (c-Fos) (b-d) Core temperature of mice treated with P57 in the MPA (b), PVH (c) or BSTLD (d). P57 (150 µg/kg) or the vehicle control was directly injected into the subregion of hypothalamus at 0 min (arrow). Core temperature (T_c) was measured and recorded by Anilogger® core

temperature monitoring system every 5 minutes. Core temperature of mice with conditional knockout PDXK or not in MPA. AAV-2/9-hSyn-EGFP-WPRE-pA or AAV2/9-hSyn-Cre-EGFP-WPRE-pA was injected into hypothalamus bilaterally for one month, then mice were treated with cold exposure (10 °C) for 24 h.

3. The authors seem unaware of the complexity of the POA circuits controlling thermogenesis as is clear from the lack of citations to important literature in this field. It seems that P57 is acting on both glutamatergic and GABAergic neurons in POA. Considering that, POA controls many homeostatic functions it is unclear from the experiments proposed whether the activation of these neurons are directly responsible for the hypothermia or they mainly control other functions (hypophagia, sleep...).

Response: Thanks for the Reviewer's comment. POA neural circuits are very complex and involved in many behaviors, such as thermogenesis, feeding, sleep, sexual behavior, and so on. In this study, we mainly focused on identifying the brain region involved in the P57-induced hypothermia, and we found the MPA is the key region. A lot of studies have reported the role of POA glutamatergic and GABAergic neurons in controlling thermoregulation^{1,2}, and the possible neural circuits have also been dissected¹⁻⁷. We are working on the subtle neural mechanism in POA underlying the hypothermia of P57 and also dissecting other possible functions of P57 in POA. We have added these comments into the Discussion in the revised manuscript (Page11, Line 257-261).

4. Line 76-79 Authors claim that motor behavior of P57-treated mice returns to normal 200 min following administration. This seems a misinterpretation of the data. First, important data about baseline motor activity are missing. Do these animals show normal motor activity before treatment? Was the experiment run at the same time of the day for each animal? light or dark phase? Second, it seems the vehicle is causing an increase in motor activity that normalizes over time, returning to the same level observed in P57-treated animals. Without baseline motor activity data, the authors cannot rule out any conclusion.

Response: Thanks for your comments. We conducted the behavior tests with appropriate control and design. For the rotarod test, we recorded the latency to fall of P57- and

vehicle-treated mice under the constant acceleration from 5 to 40 rpm within 240 seconds in light phase (Review Figure 3-3 a). For the open field test (Review Figure 3-3 b), the baseline of motor activity has been recorded before treatment, and then the motor activity was recorded at 30 minutes after intraperitoneal injection of Vehicle or P57. The motor activity was also recorded at next day when core temperature of mice recovered to normal level. We have added these results into the Figure S1d, e and the experimental details into the figure legend in the Figure S1d, e in the revised manuscript.

Review Figure 3-3 P57 reduced mouse motor behavior, which can recover to normal level. (a) Rotarod performance of P57-treated mice. (b) Locomotor behavior of P57-treated mice by Open field test.

5. Line 89- 95 Author suggest here, and in the discussion section (223- 225) that P57 has neuroprotective effects in stroke. However, a key control experiment to determine whether the neuroprotective effect is due to an effect of P57 on ischemic tissue or simply to the P57-induced hypothermia is missing. What is the effect on infarct volume if the MCAO animals were given P57, but maintained at normothermia? The discussion should reflect the lack of this important control. It is certainly interesting that P57 works in rats, a

species, like humans, that is unable to naturally express torpor. However, the mechanism of action for the induction of hypothermia could be very different between rats and mice. In this regard, many recent studies on thermoregulation highlighted the thermoregulatory mechanism differences in POA between these two species. Authors should add this caveat in the discussion and clarify why these data were obtained in a different species.

Response: Thanks for the Reviewer's comment. The Reviewer suggested that the key control experiment to determine whether the neuroprotective effect is due to an effect of P57 on ischemic tissue or simply to the P57-induced hypothermia is missing, so we conducted new experiment to evaluate the neuroprotective effects of P57 by keeping animal warm in mouse MCAO model. As shown in the following figure (Review Figure 3-4), the neuroprotective effect in mice is similar as that in rat, and keeping mice warm will abolish the neuroprotective effect of P57. We have added these results into Figure 1f and Figure S3a, b and discussed in the Discussion part in the revised manuscript.

Review Figure 3-4 P57 reduces the infarct volume in a mouse MCAO model by inducing hypothermia. P57 (25.0 mg/kg) or the vehicle was injected intraperitoneally into mice 2 h after middle cerebral artery occlusion. Core temperature of mice in MCAO + P57+ Warming group maintained by the heating pad. (a) Core temperature of mice in

MCAO model. (b) Representative images of TTC staining of brain slices in MCAO model treated with P57. (c) Statistic infarct volumes of brain slices from mouse model of MCAO.

6. Line 150. Authors stated that PL level increased in the hypothalamus but no obvious change was observed in whole brain tissues. The lack of effect could be the result of a washout effect due to a higher sampling volume. The same sample volume of other brain areas would have been a better control. In addition, why were the c-Fos data not reported for other brain areas besides the POA – did neurons in no other brain areas exhibit increased c-Fos? This would have also offered an indication of the site of action of PL or P57. The study focuses mostly on POA on the basis that other studies have found these areas to be important for induction of hypothermia. However, other thermoregulatory areas (e.g., medullary and pontine structures) should have been assessed to determine if the POA is the only site of action of P57. The experiment conducted with AAV suggests otherwise that POA is probably not the single site of action, as IP injection in POA PDXK knockout mice only reduces, instead of completely preventing, P57 induced hypothermia. The possibility of the presence of a different site of action is also supported by the high c-Fos expression in PVH, a brain area also involved in the modulation of core body temperature. Why did the authors not directly inject P57 in PVH?

Response: Thanks for the Reviewer's comment. As you suggested, we conducted new experiment to measure the PL level in the striatum, and we observed the increase tendency of PL level, but not significant (Review Figure 3-5). This result has been added into the Figure S6a in the revised manuscript.

In addition to the MPA, we also measured c-Fos expression in several other brain regions, including vPLO, PVH and BSTLD, and the c-Fos expression in PVN and BSTLD was also increased. We conducted new experiments to local microinjection of P57 into these regions and monitored mouse core temperature. Different from MPA region, P57 application in PVH and BSTLD had no obvious effect on mouse core temperature (Review Figure 3-2 b-d). Though knockout PDXK together with AAV in MPA did not induce hypothermia in housing temperature, our new result showed that reducing ambient temperature induced more temperature drop in the PDXK knockout mice (Review Figure

3-2 e). Meanwhile, our results showed that P57-induced core temperature drop was significantly reduced in the PDXK knockout mice. Therefore, P57 mainly targets the PDXK in MPA to induce hypothermia, of course we could not rule out the other region of brain and more extensive study will be needed in the future.

Review Figure 3-5 P57 significantly increased the PL level in the hypothalamus, but not in the striatum. LC-MS detection of PL in hypothalamus (left) and caudate putamen (right). Hypothalamus and caudate putamen were sampled at 90 min after treated with P57 (25.0 mg/kg).

7. Line 163 injection of P57 in POA. The volume of the injection and a clear description of the method used for injection into POA must be reported. Additionally, the size of the injection sites must be assessed from postmortem histology and plotted on atlas drawings of the mouse hypothalamus. A figure showing a histological example of the injection site and its extension into POA, for both P57 and AAV, should be added to Fig 3.

Response: Thank you for your suggestion. We have added the details of P57 injection in POA into the methods section in the revised manuscript (Page 13, Line 319-321 and Page 14, Line 331-335). The histological examples of the injection site for both P57 and AAV have been added into the Figure S6 b, f in the revised manuscript.

8. Line 198-212 None of the data obtained here are supporting the idea that “MPA is a core target of P57 for inducing hypothermia”. Keys control experiments are missing to claim this outcome.

- *Fiber photometry data: given the large injection in POA it is unclear how the authors can claim that the recorded fluorescence is coming solely from MPA neurons. Also, the placement of the recording probe is determined by the cessation of further increases in fluorescence intensity which means that if the injection x-ires-Cre was misplaced, then the recording probe was also misplaced. Fig 3e clearly shows the variability in the positions of the injection sites. Furthermore, no group data and statistical comparison are provided for these results (Fig 5f, g).*
- *Fiber optometry could produce a local temperature increase. MPA contains warm sensitive neurons that in response to warm stimulus could induce inhibition of thermogenesis and reduction in body temperature. No controls are provided for this very real possibility to explain the results.*
- *Control data of the firing rate of MPA neurons in response to the vehicle are missing. This must be provided. Also what vehicle was used?*
- *PVH shows a large C-fos activation (larger than MPA) in response to P57 administration, however, no experiments were proposed to assess the role of this nucleus which is also involved in thermoregulation (PMID: 19129373). A higher magnification photomicrograph of the PVH c-Fos results must be provided – from the low power photo, it appears that the entire PVH is c-Fos positive.*
- *“In addition, the fluctuation in the degree of GCaMP also increased with the reduction of core body temperature (Fig.5f, g and Fig.S9d, e).” how this was assessed? Group data and statistical comparison must be provided.*

Response: Thanks for the Reviewer’s comment. We have conducted several new control experiments and re-analyzed the data to support the MPA is the core target of P57 for inducing hypothermia.

For the fiber photometry experiment, although we determined the placement of the recording probe based on the fluorescence intensity during the fiber implantation, we fixed the mice to confirm the virus expression and the location of recording probe after finishing the experiment. We only used the data when the virus was expressed in MPA and recording probe targeted the MPA. We have added these details in the Methods. We

have conducted recording in several more mice, and group data and statistical comparison are also provided in the Figure 5 i,j in the revised manuscript.

We agree with the Reviewer that local temperature increase in MPA may increase the activity of warm sensitive neurons to reduce body temperature. To exclude this possibility, we conducted the new control fiber photometry recording and monitored mouse core temperature with intraperitoneal injection of P57 or vehicle, and we only observed mouse body temperature reduction with P57 injection, but not vehicle injection (Review Figure 3-6). These data have been added into the Figure S10d-g in the revised manuscript.

Review Figure 3-6 P57 injection, but not vehicle injection, reduced mouse core temperature in the fiber photometry recording experiment. Core temperature of P57-treated Vglut2-Cre mice (a) or Vgat-Cre (c). Core temperature of PL- treated Vglut2-Cre (b) or Vgat-Cre mice (d).

The 0.1% DMSO was used to dissolve P57 and PL, and we conducted new experiment to assess the effect of 0.1% DMSO on the activity of MPA neurons. As shown in Supplementary Figure 10b and c in the revised manuscript, 0.1% DMSO has no significant effect on the firing rate of MPA neurons.

P57 induced a high c-Fos expression in the PVH, and a high magnification confocal image of the PVH c-Fos expression was provided in Fig.S9a in the revised manuscript. We conducted new experiment to investigate the effect of local infusion of P57 in PVH and BSTLD on thermoregulation, and no significant effect was observed in our study (Review Figure 3-2).

The fluctuation in the degree of GCaMP was defined as the variation of GCaMP signal, and we added the measurement details in the Methods. The group data and statistical comparison are also provided in Figure 5i,j in the revised manuscript.

9. Line 219-225 “consecutive treatment of P57 maintains the lengthy hypothermic state while the thermoregulatory system remains functional when the theoretical core temperature is lowered.” It is not clear on what basis the authors can state that the thermoregulatory system remains functional. None of the proposed experiments have assessed this. “there is no tissue damage” not clear how the tissue damage was assessed. Histological sections showing ‘damage’ should be provided and a clear description should be added in the methods section.

Response: Thanks for your comment. We intended to emphasize that mouse core temperature can still back to normal after consecutive treatment with P57. As shown in the Review Figure 3-7 a, we conducted a new experiment and found that P57 induced more mouse core body temperature drop at 10°C than that at room temperature (25°C). That’s reasonable because P57 suppresses heat production with a lower metabolic rate to induce hypothermia, and the lower temperature (10°C) exacerbates the imbalance between heat production and heat dissipation. Therefore, “the consecutive treatment with P57 leaves the thermoregulatory system functional” is not appropriate, and we have deleted it in the revised manuscript.

We have added the function analyses to evaluate side-effect of P57, including heart rate and blood pressure (Review Figure 3-7 b, c). After administrated 4 consecutives i.p. of P57 to mice, heart rate firstly slowed down, then recovered. The blood pressure was hardly changed. We have added these results into Fig.S2a, b of the revised manuscript.

Review Figure 3-7 P57-induced mouse core temperature change with different ambient temperatures. (a) Mice were adapted for 2 hours at 10 $^{\circ}\text{C}$ before given P57 intraperitoneally, and then maintained another 4 hours at 10 $^{\circ}\text{C}$. 25.0 mg/kg of P57 or the vehicle control was injected intraperitoneally into mice ever 3 h for 4 times. (b) Blood pressure of P57-treated mice. (c) Heart rates of P57-treated mice.

10. Line 219-225 Based on the following statement:

- First, consecutive treatment of P57 maintains the lengthy hypothermic

- *Second, there is no tissue damage despite*
- *Third, animal is able to recover from P57-induced hypothermia*

Authors state that “Hypothermic state induced by P57 shares several features with hibernation-like state and therapeutically induced hypothermia”. I am not sure this is a good way to define a hibernation-like state, in fact, anesthetic does just the same but is not considered to induce a state of either hibernation or torpor.

Response: We agree with the Reviewer that we should cautiously claim the hibernation-like/torpor-like state, and we have changed our description in the revised manuscript.

Fig. 1 the title of the figure should reflect what is represented in the figures and on which species, in this case just the hypothermia in mice. “P57 induces a reversible hypothermia in mice”

Response: Thank you for your suggestion. I have changed figure legend as “P57 induces a reversible hypothermia in mice”.

Fig. 3 Add a panel showing an example of injection sites for AAV and P57 in POA similar Fig. 5.

Response: We have added the example panels in the revised manuscript.

Fig. 5 a panel comparing the GABA non-GABA neuronal activity in response to the vehicle should be reported. Group data for panels f & g are missing.

Response: We conducted new experiments to record MPA neuronal activity in response to the vehicle, and vehicle application had no effect on MPA neuronal activity. Results have been added into the FigureS10a in the revised manuscript. The group data for fiber photometry recording have been added in the Figure 5i,j in the revised manuscript.

Fig. S1 “Administration of P57 induces hypothermia...” however the figure does not show hypothermia but hypometabolism. The title should reflect what the figure shows.

Latency to fall, at recovery, following 200 min after P57 administration, seems extraordinarily perfect. However, from panel a it seems that, following 200 min after P57

administration, the animal has not yet recovered from the hypometabolic effect of the drugs. This seems to be true also for the locomotor activity in panel d.

Baseline locomotor activity is missing? This should be reported. Panel d seems to suggest an effect of the vehicle (maybe stress due to injection) more than a reduced motor activity induced by P57. The baseline motor activity is fundamental to show the reduction in motor activity that is claimed by the authors. Why latency to fall for the vehicle is not reported? This must be included.

Panel f. there are several informations missing that must be reported in the methods section. Time of euthanasia following injections of P57, description of the staining, and the method used to determine that there are no pathological changes (cell dimensions? Lack of presence of particular structures?).

Response: Figure legend has been revised “Administration of P57 induces hypometabolism and decreases motor activity”.

In our previous version, the latency time to fall in the rotarod test was recorded under the constant speed of 30 rpm within 240 seconds, which may make mice habituate to this test after several trials. We retested the latency time of P57 and vehicle under the constant acceleration from 5 to 40 rpm within 300 seconds, and P57 injection, but not vehicle injection, significantly reduced mouse performance, and the performance recovered after 300 min (Review Figure 3-3). We also conducted open field test again, and the baseline motor activity has been measured (Review Figure 3-3). These results have been added in the Figure S1d, e in the revised manuscript.

In panel f, mice were euthanasia after 72 hours relative to the first injection. we have added the details about this experiment in the Methods section in the revised manuscript H&E Staining.

Fig. S5 baseline motor activity should be reported in panel e. Compare the latency to fall variability between fig.S5 and S1. Why in S1 there is no variability at time 0 and time 200? Why latency to fall for the vehicle is not reported? this must be included.

Response: We have conducted both rotarod test and open field test again with PL and vehicle injection, similar as the P57 experiment (Review Figure 3-8). These results have been added in the Figure S6h, i in the revised manuscript.

Review Figure 3-8 PL reduced mouse motor behavior, which can recover to normal level. (a) Travel distance recording in open-field test at baseline before treatment with vehicle or PL, 30 minutes and 24 hours (Recovery) after intraperitoneal administration of vehicle or PL (300.0 mg/kg). (b) Rotarod performance of PL-treated mice.

Minor

In the methods section and in the relevant place in the result section, add more details on intraparenchymal injection procedures, including volume of injection, concentration, and type of vehicle used for each drug.

Response: Thank you for your advice. We have added the details about injection procedures in the methods section in the revised manuscript (Page 13-14, Line 314-339).

Line 80 should add hypophagic!! as the effects could be due to something like a fall in feeding, leading to fall in blood glucose, and maybe torpor induction.

Response: Thanks! We have added the “hypophagic” in the sentence.

Line 209 Check reference to figures. (Fig.5e-g.....) should probably be (Fig.5f-g.....).

Response: We have corrected and also checked other reference to figures.

Fig. S10 A schematic of the P57 mechanism of action should go in the main manuscript.

Response: We have moved the schematic of the P57 mechanism of action into the main manuscript.

Reviewer #4:

This is potentially interesting study, where authors identified targeted pharmacological compound P57 with ability to induce torpor or hibernation-like states in rodents. Despite important clinical significance of this topic, authors do not have enough experimental support to make this statement.

Major points:

1. Torpor is characterized by periods of complete inactivity, reduction in metabolic rate that precedes reduction in body temperature (below 31C). Behavioral experiments (rotarod and open-field) (Fig.S1) strongly indicates that authors are talking about pharmacological induction of hypothermia and hypometabolism, but NOT torpor or hibernation-like state.

Response: Thanks for your suggestion. We conducted new experiment to induce natural torpor with fasting, and found that the rate of core temperature loss was similar between fasting-induced torpor and P57 administration, but the rate of temperature gain was different (Review Figure 4-1 a). The metabolic rate is reduced by around 30% and the core temperature can reach around 31°C after intraperitoneal injection of P57 (25.0 mg/kg). Behavioral experiments (rotarod and open-field) also showed significantly reduced motor activity after P57 treatment (Review Figure 4-1 b-d; Figure 1c and Figure S1d, e of the revised manuscript). We have changed our statement and described the effects of P57 administration as inducing a hypothermia and hypometabolism in the revised manuscript to avoid the confusion as you suggested.

Review Figure 4-1 Comparison of P57-induced hypothermia with torpor or hibernation-like state. (a) Comparison of the core temperature of fasting-induced torpor (FIT) with P57-induced hypothermia. (b) Oxygen consumption level of P57-treated mice. (c) Locomotor behavior of P57-treated mice by Open field test. (d) Rotarod performance of P57-treated mice.

2. It would be useful to perform blood comprehensive metabolic panel to measure: glucose (during hibernation glucose is relatively stable); beta-hydroxybutyrate (significantly 2-3 times increased during hibernation), insulin, etc...

Response: Thank you for the valuable suggestion. We have measured blood glucose level and insulin according to the reviewer's suggestion. The blood glucose level didn't change significantly, but the insulin level reduced dramatically after intraperitoneal

administration of P57 (Review Figure 4-2). These results are similar with hibernation state^{8,9}.

Review Figure 4-2 Measurement of blood glucose and insulin of P57 treated mice.

(a) Blood glucose concentration of P57-treated mice. Blood of mice was sampled at 0 min, 60 min and 90 min after intraperitoneal injection of P57. (b) Insulin content in plasma of mice at 90 minutes after intraperitoneal injection of P57. Insulin level in plasma was detected by ELISA kit.

3. It is not clear what histological analysis (Fig. S1f) is telling us and what conclusions authors can make from it. Perhaps, it would be useful to perform staining for apoptotic markers, etc

Response: Thank you for the valuable suggestion. We have added the function analysis to enhance the conclusion, including heart rate and blood pressure. As shown in Review Figure 4-3, after administrated 4 consecutives i.p. of P57 to mice, heart rate firstly slowed down, then recovered. However, the blood pressure was hardly changed. We have added these results into Fig.S2a, b of the revised manuscript. Thus, take histological and function analysis into account, it is suggested that P57 administration did not show apparent damage after recovery. Additionally, we performed cell proliferation assay with P57 and did not observe any effect on cell growth in the presence of 50 µM P57 (data not shown).

Review Figure 4-3 Blood pressure and heart rate measurements of consecutive administration of P57. 25.0 mg/kg of P57 or the vehicle control was injected intraperitoneally into mice every 3 h for 4 times. (a) Blood pressure of P57-treated mice. (b) Heart rates of P57-treated mice.

4. It is not clear why hypothermic phenotype and mice and rats are so different upon P57 administration. In mice CBT drops to 26°C; in rats only to 34°C. One explanation is that mice have an ability to undergo natural torpor during fasting, but rats do not. Authors should discuss this species-specific phenotype. Moreover, authors conclude that “P57 can induce hibernation-like state in rodents”. I do not think that authors provide strong experimental support for this statement for mice and even less so for rats.

Response: Thank you for the thoughtful suggestions. CBT of mice drops to around 31°C, in rats only to 34°C after intraperitoneal administration of P57 once. CBT of mice drops to 26 °C after consecutively intraperitoneal administration of P57 for four times at 3-hour intervals. The specific surface area of mice is larger than rats, thus mice dissipate heat faster than rats^{10,11}, which may cause the differences of hypothermic phenotype between mice and rats.

We have changed the statement “P57 can induce hibernation-like state in rodents” with “P57 can induce hypothermia and hypometabolism in mice” according to the reviewer’s suggestion.

5. Fig. 3 Why CBT phenotype is so different from Figure 1. Max drop only to 30C. Do authors observed changes in metabolic rate similar to Fig. 1.

Response: CBT in Fig.1 were intraperitoneal temperature, which were recorded by Anilogger® core temperature monitoring system. CBT in Fig.3 were rectal temperature, which were recorded by animal temperature measuring apparatus (FT3400, Kew Basis). Thus, the difference phenotype of CBT was caused by different measurement methods.

6. Fig. S5 Behavioral data does not support the statement that animals are going into torpor or hibernation-like state. In case of mouse torpor activity is reduced almost to zero like sleep state. It is hard for me to imagine rotarod experiments during sleep.

Response: Thanks for your suggestion. We have changed our statement and described the effects of P57 administration as inducing a hypothermia and hypometabolism in the revised manuscript.

Reviewer #5:

In the manuscript “Natural Product P57 Induces Hibernation-like State through Targeting Pyridoxal Kinase” by Wang et al. the effect and function of the natural product P57 on mice and rats is monitored. The authors observe by chance that P57 administration lowers body temperature and metabolic rates. To address a potential mechanism-of-action the authors perform affinity purifications using biotinylated P57 as bait and identify PDXK as potential interactor. In in vitro analyses and structural studies, the mode of binding of P57 to PDXK is determined. P57 appears to be an inhibitor of PDXK competing with PL. Continuing with in vivo analyses the brain area in which P57 functions is determined. Using single-cell RNAseq potential target cells are identified. Finally, responses of neurons to P57 and PL in the MPA are monitored.

This is an interesting paper which spans from vitro analyses determining drug-protein interactions to in vivo studies analysing phenotypic responses of rodents to drug administration. The findings are of medical/clinical interest as induction of hibernation-like states represents an interesting treatment strategy to acute insults affecting e.g. brain or heart metabolism. However, several points should be addressed prior publication to ensure the soundness of presented data. Major points are:

(1) How were potential binding partners shortlisted in the MS-based proteomics studies? This section is kept very short and the reader cannot follow the selection criteria. It is not clear how many proteins were identified and why GSTP1 and PDXK were of specific interest.

Response: We appreciate your comments. We excised indicated bands, not the whole lanes, from gel for LC-MS/MS detection (Review Figure 5-1 a). There were 14 potential binding proteins listed in Table S1 according to the score of LC-MS/MS results between control and P57-bio group. In our study, we used the Mascot search engine to analyze the mass raw data. The mascot scoring algorithm is probability based, which can be used to judge whether the identification result is significant or not. According to mascot scoring algorithm, the score is calculated by $-10 \cdot \log_{10}(P)$, where P is the absolute probability. For example, a probability of 10^{-20} thus becomes a score of 200¹². We selected the top 3 potential binding proteins (glutathione S-transferase P1 (GSTP1), pyridoxal kinase (PDXK) and neuromodulin) for further analysis. Then affinity chromatography experiment

using mice brain lysate showed that GSTP1 and PDXK, but not neuromodulin were sensitive to competition by excess P57, thus we excluded the neuromodulin as a potential binding partner of P57 (Review Figure 5-1 b). Furthermore, the inhibitors (ethacrynic acid or celastrol) or agonist (α -angelica lactone), as well as the substrate (glutathione, GSH) of GSTP1, didn't affect mice core temperature significantly (Review Figure 5-1 c-e). However, PL, the substrate of PDXK, induced mice hypothermia. Thus, PDXK is left as a specific interest.

Review Figure 5-1 P57 interacts with PDXK. (a) Affinity chromatography experiment with mouse brain lysate using P57-Bio probe. Silver staining was performed to measure the binding proteins with P57-Bio. (b) Affinity chromatography experiment with mouse

brain lysate (c) Core temperature of EA or Cela-treated mice. (d) Core temperature of α -angelica lactone-treated mice. (e) Core temperature of GSH-treated mice.

(2) Along this line, the proteomic data should be presented “as a whole” including the shortlisted proteins, e.g. using a volcano plot, before validation experiments coupling IPs to western blots are performed.

Response: As shown in Review5 Fig. 1a, the different bands, not the whole lanes, excised from gel between control and P57-bio group were used for LC-MS/MS detection. So, we got just few hits to shortlist and validate.

(3) I did not find Table S1 accompanying the manuscript. This table is crucial to be able to evaluate proteomic data.

Response: Table S1 listed the potential proteins identified by MS of P57 pull-down assay and Table S1 is in the Supplementary materials section. We will double check the re-submitted files.

Table S1. Identified proteins by Mass spectrum of P57 pull-down assay

Protein ID	Protein name	Control		P57-biotin	
		Score	Sequence Coverage	Score	Sequence Coverage
P19157	GSTP1	0	0	584	59%
Q8K183	pyridoxal kinase	119	12%	568	39%
P06837	neuromodulin	0	0	411	66%
A6ZI44	Fructose-bisphosphate aldolase	0	0	196	25%
P35802	Neuronal membrane glycoprotein M6-a	0	0	190	17%

	Tubulin				
Q7TQD2	polymerization- promoting protein	0	0	159	35%
D3YUQ9	Elongation factor 1-delta	0	0	139	29%
P14152	Malate dehydrogenase, cytoplasmic	0	0	136	16%
P08226	Apolipoprotein E	0	0	133	16%
B1AQW2	Microtubule- associated protein	0	0	132	25%
Q8R191	Synaptogyrin-3	0	0	121	19%
B1AQW2	Microtubule- associated protein	0	0	105	18%
Q9DCJ1	Target of rapamycin complex subunit LST8	0	0	105	12%
P47754	F-actin-capping protein subunit alpha-2	0	0	100	16%

(4) In the methods section it states “Affinity chromatography experiment and iTRAQ analysis”; however, no experimental details and methods are listed that refer to the iTRAQ

labeling experiments. Thus, it is not clear if iTRAQ was used and when it was used how the ratios were treated and binders were determined.

Response: We are sorry for the mistake. The statement “Affinity chromatography experiment and iTRAQ analysis” is not accuracy. I have recorrected the statement with “Affinity chromatography experiment and LC-MS/MS analysis”.

(5) MS raw data should be deposited on a public server such as PRIDE DB.

Response: Thank you for the valuable suggestion. All the mass spectrometry data have been deposited to the iProX consortium with the dataset ID IPX0005823000 (URL: <https://www.iprox.cn/page/PSV023.html?url=1675073267636rwMF>. Password: Cd0j). We have also amended the information to the data availability in lines 784-786 of the revised manuscript.

Minor points:

(1) The manuscript should be spell- and grammar-checked by a native speaker.

(2) The introduction is very short. For non-experts it would be helpful to know why the hypothalamus is critical for body temperature control.

Response: Thank you for your suggestions. The spelling and grammar of manuscript have been carefully checked by a native speaker.

According to the reviewer’s suggestion, we have added the more information about the principle of hypothalamus for body temperature control (labeled with red color) in Introduction section of the revised manuscript.

References:

- 1 Zhao, Z. D. *et al.* A hypothalamic circuit that controls body temperature. *Proc Natl Acad Sci U S A* **114**, 2042-2047, doi:10.1073/pnas.1616255114 (2017).
- 2 Zhang, K. X. *et al.* Violet-light suppression of thermogenesis by opsin 5 hypothalamic neurons. *Nature* **585**, 420-425, doi:10.1038/s41586-020-2683-0 (2020).
- 3 Yu, S. *et al.* Glutamatergic Preoptic Area Neurons That Express Leptin Receptors Drive Temperature-Dependent Body Weight Homeostasis. *The Journal of neuroscience : the official journal of the Society for Neuroscience* **36**, 5034-5046, doi:10.1523/JNEUROSCI.0213-16.2016 (2016).
- 4 Song, K. *et al.* The TRPM2 channel is a hypothalamic heat sensor that limits fever and can drive hypothermia. *Science* **353**, 1393-1398 (2016).
- 5 Tan, C. L. *et al.* Warm-Sensitive Neurons that Control Body Temperature. *Cell* **167**, 47-59 e15, doi:10.1016/j.cell.2016.08.028 (2016).
- 6 Zhang, Z. *et al.* Estrogen-sensitive medial preoptic area neurons coordinate torpor in mice. *Nature communications* **11**, 6378, doi:10.1038/s41467-020-20050-1 (2020).
- 7 Hrvatin, S. *et al.* Neurons that regulate mouse torpor. *Nature* **583**, 115-121, doi:10.1038/s41586-020-2387-5 (2020).
- 8 Buck, M. J., Squire, T. L. & Andrews, M. T. Coordinate expression of the PDK4 gene: a means of regulating fuel selection in a hibernating mammal. *Physiol Genomics* **8**, 5-13, doi:10.1152/physiolgenomics.00076.2001 (2002).
- 9 Nelson, B. T. *et al.* Metabolic hormone FGF21 is induced in ground squirrels during hibernation but its overexpression is not sufficient to cause torpor. *PLoS One* **8**, e53574, doi:10.1371/journal.pone.0053574 (2013).
- 10 Cheung, M. C. *et al.* Body surface area prediction in normal, hypermuscular, and obese mice. *J Surg Res* **153**, 326-331, doi:10.1016/j.jss.2008.05.002 (2009).
- 11 Gilpin, D. A. Calculation of a new Meeh constant and experimental determination of burn size. *Burns* **22**, 607-611, doi:10.1016/s0305-4179(96)00064-2 (1996).
- 12 Perkins, D. N., Pappin, D. J., Creasy, D. M. & Cottrell, J. S. Probability-based protein identification by searching sequence databases using mass spectrometry data. *Electrophoresis* **20**, 3551-3567, doi:10.1002/(SICI)1522-2683(19991201)20:18<3551::AID-ELPS3551>3.0.CO;2-2 (1999).

REVIEWER COMMENTS

Reviewer #1 (Remarks to the Author):

The revised version of this manuscript has mainly addressed all my former comments. Some minor questions remain to be fixed:

1. While my overall view of the manuscript is positive, I still think that many data are infra-showed/used and they are important. For example: RQ (S1b), food intake (S1c) RQ, representative images for the MCAO setting (S3b-c), the whole Figure S10 etc. Of course, this is a likely a matter of style, but still I think those data are really important.

2. The BAT and tail IRT data (Review Figure 1) should be also added to the manuscript. However, the provide evidence show extremely reduced basal temperature levels in both areas, especially in BAT. As it can be seen in Review Figure 1a, the basal BAT temperature is $<32^{\circ}\text{C}$, which is extremely low, even for animals kept in a thermoneutral environment. This should be explained and addressed. Moreover, considering this low values, BAT UCP1 protein levels/content would provide important information. Also, in related with this, it is known that MPO effects on BAT relay in the DMH, which has not been assayed. This should be discussed.

3. Finally, although blood pressure is mainly stable, animals show a remarkable bradycardia (around 60% decrease) at 24 hours (Figure S2b). Why?

Reviewer #3 (Remarks to the Author):

The authors provided more data and satisfactory replies to some of my enquiring. Furthermore, the authors have now provided convincing evidence that the reduction of stroke is mediated by hypothermia. As I mentioned in my previous review, the new finding that P57 can induce hypothermia and hypometabolism in mice and rats (non-torpid animals) is significant, and relevant for the future therapeutic use of hypothermia in humans. They also have provided strong evidence that these thermoregulatory effects of P57 are mediated via inhibition of PDXK, resulting in an increase in brain PL, which, when injected into the POA, can also induce hypothermia. However, despite these novel findings, major concerns remain about the mechanisms by which P57 interacts with hypothalamic centers for thermogenesis. While there could be a role for MPA, the experiment design remains not adequate to exclude that other adjacent thermoregulatory areas are also involved.

1) The low (n) in the immunostaining experiment in POA may be not sufficient to discriminate which neurons are involved in the hypothermic effect of P57. vLPO and LPO might also be involved (see my comments below).

2) In the experiment in which knocking down PDXK was performed, there seems to be a large viral expression in several other thermoregulatory areas such MnPO, LPO, vLPO. How the authors can exclude the possibility that P57 injected IP did not involve all these areas? The claim is that this has been assessed by direct injection of P57 in MPA. However, the volume injected is big enough that it could have spread into adjacent areas.

3) The fiber optometry only shows a general activation of MPA glutamatergic and GABAergic neurons, and there is no evidence of whether these neurons are involved in thermoregulation or other physiological function. Same for the patch-clamp experiments. There is no characterization of the recorded neurons, to determine whether they are thermoregulatory (temperature sensitive? Insensitive? Responding to skin thermal stimuli?), if their phenotype is related to thermoregulatory function (Ex. ER α +, Q neurons), or if they project to any specific known thermoregulatory centers (DMH? RPa?).

4) The lack of complete block of the P57-induced hypothermia when knocking down PDXK could be the result of the efficiency of virus infection (as correctly pointed out by the authors in the discussion), or could be mediated by other brain areas. This has not been mentioned in their discussion. Have the

authors tried to inject P57 in MnPO, vLPO, or LPO? These are better controls than injecting into the putamen.

Major

Line 86. "A long-lasting hypothermic and hypometabolic state is similar to hibernation or daily torpor⁴¹." The citation does not support the statement. One can say that torpor is characterized by long-lasting hypothermia but having long-lasting hypothermia does not translate necessarily into being in torpor. This sentence and any reference to torpor or hibernation in the manuscript are irrelevant and should be removed. Indeed, there are no experiments presented in this manuscript designed to demonstrate whether this hypothermia or hypometabolism is a form of torpor or hibernation.

Lines 92-93. "These results indicated that P57 causes a hypophagic, hypothermic and hypometabolic state and in a reversible manner, which is similar to the hibernation-like or torpor state". This sentence should change to: "These results indicated that P57 causes a hypophagic, hypothermic and hypometabolic state in a reversible manner." None of the data presented indicate that the state induced by P57 is torpor.

Lines 154-156. "PL level, but not other vitamin B6 forms such as PN, PM and 4-PA, was dramatically increased in hypothalamus after intraperitoneal administration of P57 but there was no obvious change in whole brain or striatum tissues". Table S2 shows a significant decrease in PN level following P57 administration, but no discussion is reported about the possible role this can play in the induction of P57-mediated hypothermia.

Lines 159-161. "We thus locally administered P57 into MPA at 0.6 mg/kg and 1.2 mg/kg respectively and observed rapidly induction of deeper hypothermia in a dose-dependent manner (Fig.3b, c and Fig.S6b)". The authors report a volume of injection of 200-500nl. This is a quite big volume to inject into mice's hypothalamus. It is hard to claim selectivity on just MPA area with this large volume. Fig S6 b, I would remove the drawing of the cannula, to better show the cannula trace on the brain tissue. Add arrows to indicate the cannula trace. Still, this is not the best way to demonstrate placement for injection. The use of dye would have been more appropriate. Looking at the extension of viral diffusion (180nl injection, Ex fig. 5f, Fig S6 f), I suspect that also P57 could have diffused to a larger portion of POA. If this is the case, it is a bit of a stretch to claim that this effect is mediated exclusively by MPA.

Lines 163-165. "Moreover, we expressed Cre-EGFP in hypothalamus by injecting Pdxkflox/flox mice with AAV2/9-hSyn-Cre-EGFP-WPRE-pA, the effect of P57 (25.0 mg/kg) on inducing hypothermia was significantly reduced in the mice with knocking out PDXK in MPA neurons compared to control group (Fig. 3d~f and Fig.S6e, f)." The extension of viral diffusion goes way passed the MPA area, making it difficult to claim that this effect is mediated exclusively by MPA. It seems instead that the highest viral expression is at the periventricular level. The resolution of Fig. S6f should be increased and just a crop of the hypothalamic area should be reported. In addition, this panel should replace panel d in Fig. 3 of the manuscript.

Lines 212-213. "bed nucleus of the stria terminalis, lateral division, dorsal part (BSTLD) (Fig.S8a, b) and paraventricular nucleus of hypothalamus (PVH), but not in the vLPO (Fig.S9a~d)." The authors report no changes in c-Fos in vLPO. However, in Fig. 5 a, it is possible to appreciate quite a few cells in the vLPO area of the P57-treated animal. Moreover, there seem to be more c-Fos in LPO than in MPA in both P57- and PL-treated animals. On the contrary in Fig. S9a bottom panel, the MPA and LPO area does not seem to have lots of c-Fos, in response to either P57- or PL treatment. With such variability (compare Fig. 5a vs Fig S9a) and a low n (just 3 mice), it seems hard to obtain some useful information from this data set. Not clear from this figure if the atlas traces was used also as a counting box. If not, the author should report which area they considered for their c-Fos counts.

Lines 257-278. This entire section is mostly speculative. The only conclusion that can be drawn by the result of hypothalamic manipulation is that POA is one of the possible candidates for P57-induced. There are not sufficient pieces of evidence to claim that a specific subpopulation in POA is the main responsible for the P75-induced hypothermia.

Minor

Figures

I recommend a meticulous revision of the figures and providing a uniform style across the manuscript. Use the same nomenclature as used in the manuscript (Ex. Caudal putamen or Striatum). Use the same scale range when comparing multiple panels. The labels on the top of the panel should be consistent for all the figures. For instance, determine whether the substances injected should be reported on top of the panel (as in Fig. 3c) or within the panel legend (as in Fig. 3 e). The same logic for the site of injection, determine whether it should be reported in the legend (as in fig. 5b) or on top of the figure panel (as in Fig. S9).

Fig. 1 would be helpful for the reader to report the ambient temperature of exposure either in the figure (for each panel b, c, d, e) or in the legend. Use the same range scale for panels c, d. Panel f should contain the information on either the ambient temperature of exposure or the minimum core temperature (average \pm SEM) that each animal reached following the treatment.

Fig. 2 Label each panel in the same order from left to right. Ex panel (a) on the left should be followed by panel (b) to the right. Apply also to the other figures.

Fig. 3 Report in the legend or the figure the volume injected together with the concentration of the drug. Panel b is not informative and it should be moved to Fig. S6. Same for panel d. Then add in Fig 3 the panels b and f now present in Fig. S6.

854-855 Check this sentence in the legend "(f) was injected into hypothalamus bilaterally for one month" I believe the author meant that it was injected and let recover for 1 month.

Fig. S1 panel D. These graphs should have the same scale range. Furthermore, the author should explain why motor activity in the vehicle-treated animal is reduced compared to the baseline. In the manuscript line 84, is stated that "motor behaviors gradually returned to normal" however it appears that neither the vehicles nor P57 treated animal ever recover to baseline levels. This data should be presented in a single bar graph and analyzed with a 2-way ANOVA to show the difference between groups' treatment and time.

Fig. S3 it is not clear why the dosage of P57 is lower in rats than the one used in mice. Is there a reason for this choice? Would this have affected the reduced hypothermic effect observed in rats compared to the deep hypothermia reported in mice? Report exposure temperature in panels a, b, and c ($T_a = xx^\circ\text{C}$) or in the legend.

Fig. S6 panel figure resolution is too low to assess the extension of viral expression, it seems that viral infection affects both MPA and MnPO. The figure resolution must be increased and a high-magnification picture of just the hypothalamic area must be provided.

Reviewer #4 (Remarks to the Author):

I am satisfied with revision of the manuscript.

However, the authors used very different time frame for different experiments throughout the MS. For example, glucose was measured after 90 min post - injection, where is hypothermia was induced for up to 21 hours post injections.

Reviewer #5 (Remarks to the Author):

All concerns were adequately addressed.

Reply to Reviewer #1:

The revised version of this manuscript has mainly addressed all my former comments. Some minor questions remain to be fixed:

1. While my overall view of the manuscript is positive, I still think that many data are infra-showed/used and they are important. For example: RQ (S1b), food intake (S1c) RQ, representative images for the MCAO setting (S3b-c), the whole Figure S10 etc. Of course, this is a likely a matter of style, but still I think those data are really important.

Response: Thanks for your suggestion. We have added these figures into the main figures, such as Figure S1b and S1c into Figure 1e and 1f of the revised manuscript, representative images for the MCAO setting (S3a-b) into Figure 1i and 1j of the revised manuscript, and Figure S10a-c into Figure 5d, g and h of the revised manuscript.

2. The BAT and tail IRT data (Review Figure 1) should be also added to the manuscript. However, the provide evidence show extremely reduced basal temperature levels in both areas, especially in BAT. As it can be seen in Review Figure 1a, the basal BAT temperature is 32°C, which is extremely low, even for animals kept in a thermoneutral environment. This should be explained and addressed. Moreover, considering this low values, BAT UCP1 protein levels/content would provide important information. Also, in related with this, it is known that MPO effects on BAT relay in the DMH, which has not been assayed. This should be discussed.

Response: Thanks for your valuable suggestion. We have added Review Figure 1 into the Figure S1b and S1c of the revised manuscript. In this study, we used the IR digital thermographic camera (FLIR T430sc) to measure the body surface temperature in the BAT region of mice. The accuracy of the IR camera to measure mouse body surface temperature depends to a certain extent on the emissivity of the mice skin, imaging distance, and environmental

temperature. The emissivity of the animal skin is less than 1, so the measured temperature will be lower than the actual temperature of the mice skin surface. On the other hand, the skin surface temperature will be lower than the BAT temperature. Therefore, these reasons lead to the measured temperature being lower than the actual BAT temperature in this study. The values of temperature showed in Figure S1b and S1c are relative at a constant imaging distance and environmental temperature. The difference of temperature between vehicle and P57 ($\Delta T = T_{\text{Vehicle}} - T_{\text{P57}}$) is relative constant and more important. We added the information about temperature measurement in each figure in the Material and Methods.

We also measured the mRNA levels of UCP1 in BAT, and we did not observe significant difference (Review Figure 1-1). We have added these comments and the discussion about MPO effect on BAT relay in the DMH in Discussion section of the revised manuscript (Line 241-245).

Review Figure 1-1 BAT UCP1 mRNA levels of mice treated with P57. P57 (25 mg/kg) or the vehicle control was injected intraperitoneally into mice and UCP1 mRNA levels were detected in the BAT at 2 h after administration. Control group (1.02 ± 0.05), P57 treatment group (1.13 ± 0.11), mean ($n = 3$) \pm s.e.m.; no significant difference compared to control group, as determined by student's t test.

3. Finally, although blood pressure is mainly stable, animals show a remarkable bradycardia (around 60% decrease) at 24 hours (Figure S2b). Why?

Response: Thanks for your question. Blood pressure and heart rate are not always positively correlated one-to-one and are not automatically increase at the same time. In addition to heart rate, factors affecting blood pressure include pulse output, circulating blood volume and vascular volume, aortic and arterial wall elasticity, and peripheral resistance. Our study showed that after 24 h consecutive administration of P57, compared with the vehicle group, it did not affect the blood pressure while significantly reducing the body temperature and the heart rate of mouse, which was consistent with the relationship between body temperature and heart rate in critically ill patients observed by Broman et al, their study showed that each 1°C increase in body temperature between 32.0°C and 42.0°C was associated with an 8.35 beats/min increase in heart rate¹. It also suggests that there may be a more direct regulatory mechanism between body temperature and heart rate. We will explore this issue in our future study with our small molecules.

Reply to Reviewer #3:

The authors provided more data and satisfactory replies to some of my enquiring. Furthermore, the authors have now provided convincing evidence that the reduction of stroke is mediated by hypothermia. As I mentioned in my previous review, the new finding that P57 can induce hypothermia and hypometabolism in mice and rats (non-torpid animals) is significant, and relevant for the future therapeutic use of hypothermia in humans. They also have provided strong evidence that these thermoregulatory effects of P57 are mediated via inhibition of PDXK, resulting in an increase in brain PL, which, when injected into the POA, can also induce hypothermia. However, despite these novel findings, major concerns remain about the mechanisms by which

P57 interacts with hypothalamic centers for thermogenesis. While there could be a role for MPA, the experiment design remains not adequate to exclude that other adjacent thermoregulatory areas are also involved.

1) The low (n) in the immunostaining experiment in POA may be not sufficient to discriminate which neurons are involved in the hypothermic effect of P57. vLPO and LPO might also be involved (see my comments below).

Response: We have added 3 more data for each group in the revised manuscript. We also analyzed the vLPO and LPO subregions, and did not observe significant differences (Review Figure 3-1). We have added these data into the Figure 5b and Figure S9d and e in the revised manuscript.

Review Figure 3-1 C-Fos positive neuron counts in MPA, vLPO and LPO of P57 treatment mice. Quantification of c-Fos-positive neurons in the MPA, vLPO or LPO of each group. Dots represent the raw values in each group. Mean ($n = 6$) \pm s.e.m. Significant differences between treatments were calculated using two-way analysis of variance (ANOVA), **** $P < 0.0001$; ns, not significant.

2) In the experiment in which knocking down PDXK was performed, there seems to be a large viral expression in several other thermoregulatory areas such MnPO, LPO, vLPO. How the authors can exclude the possibility that P57

injected IP did not involve all these areas? The claim is that this has been assessed by direct injection of P57 in MPA. However, the volume injected is big enough that it could have spread into adjacent areas.

Response: Thanks for your question. Since cFos expressions in vLPO and LPO were not significantly changed, we conducted new experiment to injected 200 nL of 15 mg/mL P57 or the vehicle control into LPO, and we did not observe significant temperature change (Reviewer Figure 3-2a). We injected 200 nL P57 into the brain regions in this study, so we also checked the possible drug diffusion with same volume Dil injection, and found the Dil signal was local (Reviewer Figure 3-2b and c). Therefore, we believe that MPA in hypothalamus is a key candidate region for the role of P57 in inducing hypothermia. We have added these results in Figure S9h, k and l of revised manuscript.

Review Figure 3-2 P57 induces hypothermia mainly by working on MPA.

Core temperature of mice treated with P57 in LPO (a). P57 (150 µg/kg) or the vehicle control was injected into the LPO at 0 min (arrow). b, c, Coronal brain sections showing location of cannula implanted in MPA (b) and LPO (c), red shows the 200 nL Dil diffusion range. Mice brain was fixed at 120 min after 200 nL Dil injected into the MPA or LPO to demonstrate placement for injection and show the diffusion range of P57 or the vehicle. Scale bars, 2 mm.

3) The fiber optometry only shows a general activation of MPA glutamatergic and GABAergic neurons, and there is no evidence of whether these neurons

are involved in thermoregulation or other physiological function. Same for the patch-clamp experiments. There is no characterization of the recorded neurons, to determine whether they are thermoregulatory (temperature sensitive? Insensitive? Responding to skin thermal stimuli?), if their phenotype is related to thermoregulatory function (Ex. ER α +, Q neurons), or if they project to any specific known thermoregulatory centers (DMH? RPa?).

Response: Thank you for the constructive comments. We agree with the Reviewer about no direct evidence from our current study showing whether MPA glutamatergic and GABAergic neurons are involved in thermoregulation or not. In this study, we thus, in the Discussion section, we deliberately pointed out the known related neurons in the POA brain region that were reported to be involved in body temperature regulation (Line 267-271). We have also added the discussion about the possibility of P57 acts on MPA projecting to DMH to regulate BAT temperature (Line 241-245). *TRAP2* mice² together with optogenetic/chemogenetic tools may be a good method to further investigate the MPA neurons in P57-induced hypothermia. We are going to specifically target the P57 activated neurons in MPA to determine their role in thermoregulation in our follow-up work.

4) The lack of complete block of the P57-induced hypothermia when knocking down PDXK could be the result of the efficiency of virus infection (as correctly pointed out by the authors in the discussion), or could be mediated by other brain areas. This has not been mentioned in their discussion. Have the authors tried to inject P57 in MnPO, vLPO, or LPO? These are better controls than injecting into the putamen.

Response: Thanks for your valuable suggestion. We conducted new experiment to inject P57 into LPO and did not observe temperature change (Reviewer Figure 3-2a, Figure S9h of the revised manuscript). Other brain regions, such as DMH, may also be involved in the process of P57-induced hypothermia, which needs to be further explored in future. We have added

these statements into the Discussion in the revised manuscript (Line 241-245).

Major

Line 86. "A long-lasting hypothermic and hypometabolic state is similar to hibernation or daily torpor⁴¹." The citation does not support the statement. One can say that torpor is characterized by long-lasting hypothermia but having long-lasting hypothermia does not translate necessarily into being in torpor. This sentence and any reference to torpor or hibernation in the manuscript are irrelevant and should be removed. Indeed, there are no experiments presented in this manuscript designed to demonstrate whether this hypothermia or hypometabolism is a form of torpor or hibernation.

Response: Thanks for your comment. We have deleted this sentence and the reference.

Lines 92-93. "These results indicated that P57 causes a hypophagic, hypothermic and hypometabolic state and in a reversible manner, which is similar to the hibernation-like or torpor state". This sentence should change to: "These results indicated that P57 causes a hypophagic, hypothermic and hypometabolic state in a reversible manner." None of the data presented indicate that the state induced by P57 is torpor.

Response: Thanks, we have changed the sentence as you suggested.

Lines 154-156. "PL level, but not other vitamin B6 forms such as PN, PM and 4-PA, was dramatically increased in hypothalamus after intraperitoneal administration of P57 but there was no obvious change in whole brain or striatum tissues". Table S2 shows a significant decrease in PN level following P57 administration, but no discussion is reported about the possible role this can play in the induction of P57-mediated hypothermia.

Response: Thanks for your question. In this study, mass spectrometry was used to analyze the changes of B6 vitamers in the whole brain, hypothalamus

and striatum of mice after intraperitoneal injection of P57, and it was found that only the level of PL increased significantly in the hypothalamus, while the level of PN decreased significantly in the whole brain (Table S2). However, there was no significant change in striatum (Review Figure 3-3a and b). By intraperitoneal injection of 300 mg/kg PL, PN, PM and PLP, we found that only PL induced hypothermia in mice. These results further demonstrate that PL is a key B6 vitamer during P57-induced hypothermia.

As for the significant reduction of PN in whole brain after P57 treatment, we speculated that it was to maintain the stability of PLP content in cells to ensure that cells could carry out normal life activities, because PLP is a coenzyme of more than 160 enzymes. Review Figure 3-3c shows the metabolic and transformation process of B6 vitamers³. According to our conclusion, P57 inhibits PDXK enzyme activity. On the basis of Review Figure 3-3c, PLP content should be reduced, but it can be seen in the mass spectrum results of Table 2 and 3 that PLP content does not change significantly after P57 treatment. Therefore, we speculated that when P57 inhibits PDXK enzyme activity, it may accelerate the conversion of PNP into PLP to maintain the relative stability of PLP content and enable cells to maintain normal life activities. When a large amount of PNP is converted into PLP, the conversion process of PNP to PN becomes less, which ultimately leads to the reduction of PN. Similar to PN, this may also be the reason why PM content decreased after P57 treatment for 90 min (Table S2 of the revised manuscript).

Review Figure 3-3 Concentration of PN in hypothalamus and striatum and B6 vitamers metabolism. a, b, LC-MS detection of PN in hypothalamus (a) and striatum (b). Hypothalamus and striatum were sampled at 90 min after treated with P57 (25.0 mg/kg), mean (n = 9) ± s.e.m.; no significant difference compared to control group, as determined by student's t test. Detection of PN in tissues was confirmed by mass and retention time relative to the pure standard. c, the schematic metabolism of vitamin B6³.

Lines 159-161. "We thus locally administered P57 into MPA at 0.6 mg/kg and 1.2 mg/kg respectively and observed rapid induction of deeper hypothermia in a dose-dependent manner (Fig.3b, c and Fig.S6b)". The authors report a volume of injection of 200-500nl. This is a quite big volume to inject into mice's hypothalamus. It is hard to claim selectivity on just MPA area with this large volume. Fig S6 b, I would remove the drawing of the cannula, to better show the cannula trace on the brain tissue. Add arrows to indicate the cannula trace. Still, this is not the best way to demonstrate placement for injection. The use of dye would have been more appropriate. Looking at the extension of viral diffusion (180nl injection, Ex fig. 5f, Fig S6 f), I suspect that also P57 could have diffused to a larger portion of POA. If this is the case, it is a bit of a stretch to claim that this effect is mediated exclusively by MPA.

Response: Thanks for your suggestion. We have change the Figure S6 and now it is Figure 3b in the revised manuscript. we conducted new experiment to injected 200 nL of 15 mg/mL P57 (150 µg/kg) or the vehicle control into LPO, and we did not observe significant temperature change (Reviewer Figure 3-2a). We injected 200 nL P57 into the brain regions in this study, so we checked the possible drug diffusion with same volume Dil injection, and found the Dil signal was local (Reviewer Figure 3-2b and c). Therefore, we believe that MPA in hypothalamus is a key candidate region for the role of P57 in inducing hypothermia. We have added these results in Figure S9h, k and l of revised manuscript.

Lines 163-165. “Moreover, we expressed Cre-EGFP in hypothalamus by injecting *Pdxkflox/flox* mice with AAV2/9-hSyn-Cre-EGFP-WPRE-pA, the effect of P57 (25.0 mg/kg) on inducing hypothermia was significantly reduced in the mice with knocking out PDXK in MPA neurons compared to control group (Fig. 3d~f and Fig.S6e, f).” The extension of viral diffusion goes way passed the MPA area, making it difficult to claim that this effect is mediated exclusively by MPA. It seems instead that the highest viral expression is at the periventricular level. The resolution of Fig. S6f should be increased and just a crop of the hypothalamic area should be reported. In addition, this panel should replace panel d in Fig. 3 of the manuscript.

Response: Yes, due to the extension of viral diffusion goes way passed the MPA area, we add the claim that “we cannot rule out the possibility that P57 acts in other brain regions to induce hypothermia” in Discussion section (Line 241-242). We also provided the high resolution (Review Figure 3-4a) and added the high magnification images (Review Figure 3-4b) of the hypothalamic area into the revised manuscript.

Review Figure 3-4 AAV2/9-hSyn-EGFP or AAV2/9-hSyn-Cre-EGFP expressed in MPA. a, b, Coronal brain sections showing EGFP (upper) or Cre-EGFP (lower) expressed in MPA.

Lines 212-213. “bed nucleus of the stria terminalis, lateral division, dorsal part (BSTLD) (Fig.S8a, b) and paraventricular nucleus of hypothalamus (PVH), but not in the vLPO (Fig.S9a~d).” The authors report no changes in c-Fos in vLPO. However, in Fig. 5 a, it is possible to appreciate quite a few cells in the vLPO area of the P57-treated animal. Moreover, there seem to be more c-Fos in LPO than in MPA in both P57- and PL-treated animals. On the contrary in Fig. S9a bottom panel, the MPA and LPO area does not seem to have lots of c-Fos, in response to either P57- or PL treatment. With such variability (compare Fig. 5a vs Fig S9a) and a low n (just 3 mice), it seems hard to obtain some useful information from this data set. Not clear from this figure if the atlas traces was used also as a counting box. If not, the author should report which area they considered for their c-Fos counts.

Response: We appreciate for your suggestion. We have added 3 more data for each group in the revised manuscript. We also analyzed the vLPO and LPO subregions, and did not observe significant differences (Review Figure 3-1). We have added these data into the Figure 5b and Figure S9d and e in the revised manuscript. We replaced the new example diagram of Figure 5a (Review Figure 3-5). We aligned brain sections with their corresponding brain areas in the Allen brain atlas, and then the numbers of c-FOS positive cells in each brain area were counted. We have added this information into the Methods in the revised manuscript

Review Figure 3-5 P57 and PL activate neurons in MPA. Brain sections containing POA immunostained with a neuronal activation marker (c-Fos) 160 min after intraperitoneal injection of vehicle, P57 (25.0 mg/kg) or PL (300.0 mg/kg) (representative of n = 6 mice). Scale bars, 100 μ m.

Lines 257-278. This entire section is mostly speculative. The only conclusion that can be drawn by the result of hypothalamic manipulation is that POA is one of the possible candidates for P57-induced. There are not sufficient pieces of evidence to claim that a specific subpopulation in POA is the main responsible for the P75-induced hypothermia.

Response: Yes, our study does not have direct evidence to claim a specific subpopulation in MPA is the main responsible for P57-induced hypothermia, but our results support the MPA is the key target of P57 to induce hypothermia. We explored the possible target neurons of P57 in MPA, and found both MPA glutamatergic and GABAergic neurons can be activated by P57. Further study combining TRAP mouse line and optogenetic/chemogenetic tools may help us dissect the specific MPA neuron and neural circuit involved in the P57-induced hypothermia. We are designing experiments to explore the specific MPA subpopulation involved in P57-induced hypothermia in our next story. These statements have been added into the Discussion in the revised manuscript.

Minor

Figures

I recommend a meticulous revision of the figures and providing a uniform style across the manuscript. Use the same nomenclature as used in the manuscript (Ex. Caudal putamen or Striatum). Use the same scale range when comparing multiple panels. The labels on the top of the panel should be consistent for all the figures. For instance, determine whether the substances injected should be reported on top of the panel (as in Fig. 3c) or within the panel legend (as in Fig. 3 e). The same logic for the site of injection, determine whether it should be reported in the legend (as in fig. 5b) or on top of the figure panel (as in Fig. S9).

Response: Thanks for your suggestion. We have changed all the caudal putamen into striatum. The substances injected were reported within the panel legend (Figure 1b-e, g and i, Figure 3c, e, f and h-j, Figure 5c, Figure S1a-c, Figure S3a, Figure S4c-e, Figure S6c, f, Figure S9f-h, Figure S10). The sites of injection were reported on the top of the figure panel (Figure 3c and i, Figure 5b and c, Figure S6c, Figure S9b-l).

Fig. 1 would be helpful for the reader to report the ambient temperature of exposure either in the figure (for each panel b, c, d, e) or in the legend. Use the same range scale for panels c, d. Panel f should contain the information on either the ambient temperature of exposure or the minimum core temperature (average \pm SEM) that each animal reached following the treatment.

Response: Thanks for your valuable suggestion. We have added the ambient temperature of exposure in the figure legend for Figure 1. We have replaced the Figure 1c and d to ensure that their range scale is the same. We also added the Figure 1i of the revised manuscript to show the information of the mice's core temperature following the treatment.

Fig. 2 Label each panel in the same order from left to right. Ex panel (a) on the left should be followed by panel (b) to the right. Apply also to the other figures.

Response: Thanks. We have adjusted these labels (Figure 2b and c, Figure 4e and f, Figure S4b and c).

Fig. 3 Report in the legend or the figure the volume injected together with the concentration of the drug. Panel b is not informative and it should be moved to Fig. S6. Same for panel d. Then add in Fig 3 the panels b and f now present in Fig. S6.

Response: Thanks for your suggestion. We have added the volume and concentration information of the drug in Figure 3. We have added the old Figure S6b to the present Figure 3b and added the high resolution and high magnification images (Review Figure 3-4b) of a crop of the hypothalamic area to Figure 3d. We also delete the low-resolution version of Figure S6f.

854-855 Check this sentence in the legend “(f) was injected into hypothalamus bilaterally for one month” I believe the author meant that it was injected and let recover for 1 month.

Response: You are right, and we have corrected this statement.

Fig. S1 panel D. These graphs should have the same scale range. Furthermore, the author should explain why motor activity in the vehicle-treated animal is reduced compared to the baseline. In the manuscript line 84, is stated that “motor behaviors gradually returned to normal” however it appears that neither the vehicles nor P57 treated animal ever recover to baseline levels. This data should be presented in a single bar graph and analyzed with a 2-way ANOVA to show the difference between groups' treatment and time.

Response: We appreciate for your suggestion. We put this data together in a single bar graph and analyzed with a 2-way ANOVA to show the difference between groups' treatment and time (Figure S1d of the revised manuscript). The motor activity was measured 24 h later, and animals may also adapt to the environment and decrease the motor activity. We have changed the statement into “motor behaviors can gradually recover”.

Fig. S3 it is not clear why the dosage of P57 is lower in rats than the one used in mice. Is there a reason for this choice? Would this have affected the reduced hypothermic effect observed in rats compared to the deep hypothermia reported in mice? Report exposure temperature in panels a, b, and c ($T_a = xx^\circ\text{C}$) or in the legend.

Response: Thanks for your question. Different species have different tolerances to the same drug, and the dosage between different species is often converted as mg/m² according to a study by pinkle in 1958⁴. Therefore, based on the Meeh-Rubner formula converted from body surface area, we calculated that the dose of 25 mg/kg (or 12.5 mg/kg) used in mice was equivalent to the dose of 17.5 mg/kg (or 8.75 mg/kg) used in rats, and we used this dose in rats. According to the effects of different doses of P57 on the body temperature of mice and rats, we can speculate that the different dosage will affect the reduced hypothermic effect observed in rats. In this work, we chose mice as our core experimental subjects for a large number of studies, and did not repeat the same on rats. We have added the ambient temperature of exposure in the figure legend for Figure S3 of the revised manuscript.

Fig. S6 panel figure resolution is too low to assess the extension of viral expression, it seems that viral infection affects both MPA and MnPO. The figure resolution must be increased and a high-magnification picture of just the hypothalamic area must be provided.

Response: We appreciate for your suggestion. Due to the extension of viral diffusion goes way passed the MPA area, we add the claim that “we cannot rule out the possibility that P57 acts in other brain regions to induce hypothermia” in Discussion section (Line 241-242). The results of cannula administration in different brain areas lead us to believe that MPA in hypothalamus is a key candidate region for the role of P57 in inducing hypothermia. We have provided the high resolution (Review Figure 3-4a) and added high magnification images

(Review Figure 3-4b) into Figure 3d in the revised manuscript.

Reply to Reviewer #4:

I am satisfied with revision of the manuscript.

However, the authors used very different time frame for different experiments throughout the MS. For example, glucose was measured after 90 min post - injection, where is hypothermia was induced for up to 21 hours post injections.

Response: Thanks for your suggestion. In our study, one-time P57 injection (25 mg/kg) induced hypothermia for ~300 min and the hypothermia reached minimum value around 120 min (Figure 1b), therefore, the duration of the main temperature measurement experiments reached 300 min after administration. And we measured the PL level and cFos expression between 90-120 min. Four times P57 injection could induce hypothermia for more than 30 hours (Figure 1g), and we mainly focused on the effect of one-time P57 injection in this study.

Reply to Reviewer #5:

All concerns were adequately addressed.

Response: Thanks for your valuable advice to help us improve the quality of our work!

Reference

- 1 Broman, M. E., Vincent, J. L., Ronco, C., Hansson, F. & Bell, M. The Relationship Between Heart Rate and Body Temperature in Critically Ill Patients. *Crit Care Med* **49**, e327-e331, doi:10.1097/CCM.0000000000004807 (2021).
- 2 Allen, W. E. *et al.* Thirst-associated preoptic neurons encode an aversive motivational drive. *Science* **357**, 1149-1155, doi:10.1126/science.aan6747 (2017).
- 3 Galluzzi, L. *et al.* Effects of vitamin B6 metabolism on oncogenesis, tumor progression and therapeutic responses. *Oncogene* **32**, 4995-5004, doi:10.1038/onc.2012.623 (2013).
- 4 Pinkel, D. The use of body surface area as a criterion of drug dosage in cancer chemotherapy. *Cancer Res* **18**, 853-856 (1958).

REVIEWERS' COMMENTS

Reviewer #3 (Remarks to the Author):

The authors have replied appropriately to many of my previous comments. Major concerns remain on MPA's role.

As previously mentioned the injections performed at the hypothalamic areas were quite large in volume as clearly demonstrated by the lesion appearing at the injection site (Fig. S9). It is a bit hard to claim whether the effect in MPA is due to the drug or the lesion. However, in response to my previous comment, authors have provided a higher "n" for the Fos study in MPA that coupled with the conditional knockout PDXK in MPA (although viral diffusion goes way passed the MPA area as admitted by the authors) allow to "suggest" not "prove" that MPA might be involved. Patch-clamp recording and fiber optometry remain problematic (see comments below). In my opinion, the authors do not have strong evidence to prove that MPA is the key target to mediate P57-induced hypothermia. Hence any claim in the result or the discussion sections about MPA's role as a key target to mediate P57-induced hypothermia should be tuned down or removed. Any sentence on the role of MPA should sound "suggestive" or "speculative". The caveat of the lesion at the injection site should be reported as possibly confounding for the interpretation of the data.

Line 239 "Since knocking down PDXK in MPA..." knocking down was done in a much large area of the hypothalamus that includes MPA. This sentence should change to report the reality of what has been done.

Line 236 "local microinjection of P57 into MPA....." and lesion of the area.

Lines 232- 242 this al paragraph should be reworked to reduce the emphasis on the MPA role especially if the data on electrophysiological recording and fiber photometry recording are not reaching statistical significance.

This paragraph should sound like: We further investigate the possible targets of P57 in the central nervous system, and found that the preoptic area of the hypothalamus is a key candidate region. Based on previous literature[cit] and some of our results we speculate that MPA could play a key role in However,.....add here the caveats of the results.....[e.g. big injection, Lesions, extension of knocking down]

Line 267 "In our study, we identify the POA, especially the MPA nucleus, is a key brain target region" This sentence should change to "In our study, we identify that POA, is a key brain target region of P57 to induce hypothermia.

Line 218-219 "We then directly recorded the activity of MPA neurons in slice with the patch-clamp technique. Either P57 or PL application increased the firing rate of MPA neurons (Fig.5d-h)." Is there any statistical significance that proves this increase? If yes, indicate statistical significance with an asterisk in the relative panel of Figure 5, or report this info in the legend. If not significantly different, remove this sentence from the results and any relative claim from the discussion. In this case, the figures should be either removed or moved into the supplementary with a clear description that non-significant change was found in response to..... The same reasoning for the fiber photometry results. See next comment.

Line 222-227 "GCaMP fluorescence signal from both GABA and glutamate neurons was increased with the falling of body temperature after intraperitoneal administration of P57 or PL (Fig.5j-m and Fig.S10a-d). In addition, the fluctuation degree of GCaMP also increased with the reduction of core body temperature (Fig.5j, k and Fig.S10a-d)." any statistical significance that proves these changes? If yes, indicate statistical significance with an asterisk in the relative panel of Figure 5, or report this info in the legend. If not significantly different, remove this sentence from the results and any relative claim from the discussion. Figures can be removed or moved into supplementary data (as for the comment above).

Line 161-164 "We thus locally administered P57 into MPA at 0.6 mg/kg and 1.2 mg/kg respectively and observed rapidly induction of deeper hypothermia in a dose-dependent manner (Fig.3b, c). However, the core temperature of mice didn't change when P57 was administrated directly into striatum (Fig.S6b, c)."

Control injections were also done into PVH, BSTLD, LPO, VLPO. Add this info and reference to the relative figure here.

Reply to Reviewer #3:

The authors have replied appropriately to many of my previous comments.

Major concerns remain on MPA's role.

As previously mentioned the injections performed at the hypothalamic areas were quite large in volume as clearly demonstrated by the lesion appearing at the injection site (Fig. S9). It is a bit hard to claim whether the effect in MPA is due to the drug or the lesion. However, in response to my previous comment, authors have provided a higher “n” for the Fos study in MPA that coupled with the conditional knockout PDXK in MPA (although viral diffusion goes way passed the MPA area as admitted by the authors) allow to “suggest” not “prove” that MPA might be involved. Patch-clamp recording and fiber optometry remain problematic (see comments below). In my opinion, the authors do not have strong evidence to prove that MPA is the key target to mediate P57-induced hypothermia. Hence any claim in the result or the discussion sections about MPA's role as a key target to mediate P57-induced hypothermia should be tuned down or removed. Any sentence on the role of MPA should sound “suggestive” or “speculative”. The caveat of the lesion at the injection site should be reported as possibly confounding for the interpretation of the data.

Response: Thank you for the constructive comments. We add to the discussion that lesions in this region may affect P57 induced hypothermia (Line 249). In addition, we have tuned down the statement about the role of MPA as the key target of P57 to induce hypothermia and moved the data about MPA into the supplementary figure in the revised manuscript.

Line 239 “Since knocking down PDXK in MPA....” knocking down was done in a much large area of the hypothalamus that includes MPA. This sentence should change to report the reality of what has been done.

Response: Thanks for your comment. We have changed the sentence as you suggested (Line 250-252).

Line 236 “local microinjection of P57 into MPA.....” and lesion of the area.

Response: Thanks for your suggestion. We add to the discussion that lesions in this region may affect P57 induced hypothermia (Line 249).

Lines 232- 242 this al paragraph should be reworked to reduce the emphasis on the MPA role especially if the data on electrophysiological recording and fiber photometry recording are not reaching statistical significance.

*This paragraph should sound like: We further investigate the possible targets of P57 in the central nervous system, and found that the preoptic area of the hypothalamus is a key candidate region. Based on previous literature[*cit*] and some of our results we speculate that MPA could play a key role in However,add here the caveats of the results.....[e.g. big injection, Lesions, extension of knocking down].*

Response: Thank you for the constructive comments. We have recalibrated this paragraph to reduce the emphasis on MPA and to state the possible effects of injection doses, tissue lesions, and virus spread.

Line 267 “In our study, we identify the POA, especially the MPA nucleus, is a key brain target region” This sentence should change to “In our study, we identify that POA, is a key brain target region of P57 to induce hypothermia.

Response: Thanks, we have changed the sentence as you suggested.

Line 218-219 “We then directly recorded the activity of MPA neurons in slice with the patch-clamp technique. Either P57 or PL application increased the firing rate of MPA neurons (Fig.5d-h).” Is there any statistical significance that proves this increase? If yes, indicate statistical significance with an asterisk in the relative panel of Figure 5, or report this info in the legend. If not significantly different, remove this sentence from the results and any relative claim from the discussion. In this case, the figures should be either removed

or moved into the supplementary with a clear description that non-significant change was found in response to..... The same reasoning for the fiber photometry results. See next comment.

Response: Thanks for your valuable suggestion. We analyzed Fig.5g and h for statistical differences and found that P57 treatment significantly increased the activity of GABA neurons in the MPA, which was marked in the relative panel. At the same time, we also move Fig.5d-h to the supplementary (Fig.S10a-e of the revised manuscript).

Line 222-227 “GCaMP fluorescence signal from both GABA and glutamate neurons was increased with the falling of body temperature after intraperitoneal administration of P57 or PL (Fig.5j-m and Fig.S10a-d). In addition, the fluctuation degree of GCaMP also increased with the reduction of core body temperature (Fig.5j, k and Fig.S10a-d).” any statistical significance that proves these changes? If yes, indicate statistical significance with an asterisk in the relative panel of Figure 5, or report this info in the legend. If not significantly different, remove this sentence from the results and any relative claim from the discussion. Figures can be removed or moved into supplementary data (as for the comment above).

Response: Thanks for your valuable suggestion. We analyzed Fig.5i and j for statistical differences and found that both P57 and PL treatment significantly increased the activity of glutamate neurons in the MPA, which was marked in the relative panel. At the same time, we also move these figures to the supplementary (Fig.S10f-n of the revised manuscript).

Line 161-164 “We thus locally administered P57 into MPA at 0.6 mg/kg and 1.2 mg/kg respectively and observed rapidly induction of deeper hypothermia in a dose-dependent manner (Fig.3b, c). However, the core temperature of mice didn’t change when P57 was administrated directly into striatum (Fig.S6b, c).”

Control injections were also done into PVH, BSTLD, LPO, VLPO. Add this info and reference to the relative figure here.

Response: Thanks for your suggestion. We have added this info in the revised manuscript (Line 171-173).